

# Modeling forest evapotranspiration and water balance at stand and catchment scales: a spatial approach

Samuli Launiainen[1], Mingfu Guan[2,1], Aura Salmivaara[1], and Antti-Jussi Kieloaho[1]

[1]Nature Resources Institute Finland, Latokartanonkaari 9, 00790 Helsinki, Finland
[2]Department of Civil Engineering, The University of Hong Kong, HKSAR, China

**Correspondence:** Samuli Launiainen (samuli.launiainen@luke.fi)

**Abstract.**

Vegetation is known to have strong influence on evapotranspiration (ET), a major component of terrestrial water balance. Yet hydrological models often describe ET by methods unable to sufficiently include the variability of vegetation characteristics in their predictions. To take advantage of increasing availability of high-resolution open GIS-data on land use, vegetation and soil characteristics in the boreal zone, a modular, spatially distributed model for upscaling ET and other hydrological processes from a grid cell to a catchment level is presented and validated. An improved approach to upscale stomatal conductance to canopy scale using information on plant type (conifer / deciduous) and stand leaf-area index (LAI) is proposed by coupling a common leaf-scale stomatal conductance model with a simple canopy radiation transfer scheme. Further, a generic parametrization for vegetation and snow-related hydrological processes for Nordic boreal forests is derived based on literature and data from a boreal FluxNet site. With the generic parametrization, the model was shown to well reproduce daily ET measured by eddy-covariance technique at ten conifer-dominated Nordic forests whose LAI ranged from 0.2 to 6.8 $m^2m^{-2}$. Topography, soil and vegetation properties at 21 small boreal headwater catchments in Finland were derived from open GIS-data at 16 x 16 m grid size to upscale water balance from a stand to catchment level. The predictions of annual ET and specific discharge were successful in all catchments, located from 60 to 68 °N, and daily discharge also reasonably well predicted by calibrating only one parameter against discharge data measurements. The role of vegetation heterogeneity on soil moisture and partitioning of ET was demonstrated. The proposed approach can support e.g. in forest trafficability forecasting and predicting the impacts of climate change and forest management on stand and catchment water balance. With appropriate parametrization it can be generalized outside the boreal coniferous forests.

# 1 Introduction

The boreal region, encompassing ca. 12 % of world's land area, is characterized by mosaic of peatlands, lakes and forests of different ages and structures. The landscape heterogeneity has major influence on hydrological cycle, carbon balance and



land-atmosphere interactions in the region (McDonnell et al., 2007; Govind et al., 2011; Stoy et al., 2013; Chapin Iii et al., 2000; McGuire et al., 2002; Karlsen et al., 2016). Understanding spatial and temporal patterns in hydrological fluxes and state variables is becoming increasingly important in the context of intensifying use of boreal forests under the pressures from climate change (Bonan, 2008; Gauthier et al., 2015; Price et al., 2013; Spittlehouse, 2005; Laudon et al., 2016). Thus,
model approaches that can effectively utilize available environmental data, open high-resolution GIS-data and remote-sensing products in hydrological predictions are necessary for climate-smart and environmentally sustainable use of boreal ecosystems (Mendoza et al., 2002).

Numerous models for predicting point scale, catchment level and regional hydrological balance have been developed. They range from lumped rainfall-runoff schemes (Beven and Kirkby, 1979; Bergström, 1992) to semi- and fully distributed,
physically-based models (Vivoni et al., 2011; Panday and Huyakorn, 2004; Ivanov et al., 2004; Ma et al., 2016; Clark et al., 2015a, b; Best et al., 2011; Niu et al., 2011; Ala-aho et al., 2017). Lumped models are often based on conceptual representation of hydrological processes and their parametrization is seldom based on measurable physical properties. They rely on calibration against a few integrative measures, such as the stream discharge, which is problematic for un-gauged catchments where such data is not available. Another drawback of lumped models is that they cannot address spatial variability of hydrological fluxes
and state variables. Distributed models, on the other hand, use first principles to predict water flows and state variables through the landscape and can thus incorporate topographic variability, soil texture and vegetation heterogeneity in their predictions (Samaniego et al., 2010). However, high computational costs and challenges to estimate spatially variable parameters hampers their use and performance over large areas (Freeze and Harlan, 1969; Montanari and Koutsoyiannis, 2012; Grayson et al., 1992; Reed et al., 2004).

The availability of good-quality high-resolution open data on land use, topography, vegetation and soil characteristics has increased significantly in the recent decade. In Finland, for instance high-accuracy digital elevation models (DEM's) are openly available at 2m and 10m resolution (NSLF, 2017), reasonably good soil maps cover the country at scale of 1:20 000 or 1:200 000 (GSF, 2015), and the multi-source National Forest Inventory (mNFI, Mäkisara et al. (2016); Kangas et al. (2018)) provides information on various forest and site type attributes at 16m resolution throughout the country. To take full advantage of open
GIS data and fine-resolution (i.e. tens to hundreds of meters) remote-sensing products of hydrological fluxes and state variables (Ryu et al., 2011; Herman et al., 2018), computationally efficient models capable of accounting for the landscape variability are necessary. Further, these models should be sufficiently generic in their parametrization and use standard meteorological data to allow supporting their use on large, often data-sparce areas.

Within a specific biome and climatic region, vegetation characteristics such as species composition and leaf-area index (LAI)
have major influence on variability of evapotranspiration (ET) (Williams et al., 2012; Zeng et al., 2018; Launiainen et al., 2016). At present, empirical and simple approaches are commonly used to simulate ET in hydrological models. These include computing the bulk ET as a crop or vegetation type dependent fraction of potential evaporation, or using Penman-Monteith equation (Zhao et al., 2013; Fisher et al., 2005). In both approaches, the non-linear dependency of ET (and its components) on LAI and vegetation composition remain mostly unresolved and need to be imposed through model parameterization. Improving
ET description by more physiologically-phased approach can be proposed as one potential area to reduce uncertainties in




streamflow and soil moisture predictions. Moreover, as open spatial data on vegetation and its functioning is getting more and more accurate, description of ET needs to be adapted to take full advance of such information. In modern Land Surface Models (LSM's) ET components are computed either using a big-leaf framework or by describing the microclimatic gradients and exchange rates explicitly throughout a multi-layer canopy-soil system and upscaling these directly to ecosystem scale (Katul

et al., 2012; Bonan et al., 2014). In both cases the upscaling of stomatal conductance $g_s$ and transpiration rate from leaf to canopy scale is based on physical arguments, and constrained by plant carbon economy (Cowan and Farquhar, 1977; Katul et al., 2012; Medlyn et al., 2012) and hydraulic architecture (Sperry, 2000; Tyree and Zimmermann, 2002).

Motivated both by scientific needs and potential practical applications, this study addresses two independent but inter-related goals: First, we explore whether information of LAI and plant physiologic traits can be used to derive a model for canopy

conductance $G_c$, the stomatal conductance integrated of the canopy depth. We hypothesize that coupling the unified stomatal model (Medlyn et al., 2012) with the leaf-to-canopy upscaling based on simplified canopy radiative transfer theory (Saugier and Katerji, 1991; Kelliher et al., 1995; Leuning et al., 2008) can provide means to obtain defensible estimate for $G_c$. Second, we extend stand scale predictions into a catchment level by developing a computationally efficient semi-distributed model taking advance of open GIS-data. We expect the vegetation hydrological processes be sufficiently generic to allow upscaling

of water budgets from a grid cell scale to a catchment level using a generic parametrization.

To reach these goals, we first describe the Spatial Forest Hydrology model (SpaFHy) that couples a fully distributed representation of above-ground and topsoil hydrology, controlled by vegetation and soil characteristics, with a topography-driven hillslope/catchment model. The model predictions of ET are first validated at stand scale against eddy-covariance (EC) measurements from ten boreal forest and peatland sites in Finland and Sweden (Launiainen et al., 2016), and complemented by

parameter sensitivity analysis. The soil moisture and snow predictions are compared at one site. We then proceed to a larger scale and apply SpaFHy to 21 headwater catchments located throughout Finland to validate its predictions against daily stream discharge and annual ET derived from catchment water balance. The spatial predictions of ET, soil moisture and temporal variability of stream discharge are then demonstrated at a single catchment and potential applications finally discussed.

## 2   Model description

SpaFHy consists of three sub-models (Fig. 1. Hydrological processes in vegetation and two-layer topsoil are explicitly modelled for each grid cell, while Topmodel concept (Beven and Kirkby, 1979) is used to link grid cell and catchment water budgets, and to describe the baseflow and returnflow generation mechanisms. The SpaFHy submodels and equations are presented in the next sections, and complemented in Supplementary material. Throughout, we use notation where angle parenthesis $\langle y \rangle$ indicate spatial and $\overline{y}$ temporal averages of quantity $y$.

### 30  2.1   Canopy sub-model: above-ground water budget and fluxes at a grid cell

The hydrological processes in vegetation canopy, forest floor, snowpack, and in organic moss/humus layer and underlying root zone are solved for each grid cell using information on stand structure and soil type (Fig. 1; *Canopy* and *Bucket* -submodels).





### 2.1.1 P-M equation and ET

Total evapotranspiration is defined as sum of physiologically controlled transpiration ($T_r$) and physically regulated evaporation from wet canopy ($E$) and forest floor ($E_f$). To account for different controls of these processes, a three-source model is applied to describe ET at a grid-cell scale (Fig. 1). The Penman-Monteith (hereafter referred as P-M)) equation gives each component of ET (mm d$^{-1}$) as (Monteith and Unsworth, 2008)

$$E_i = \frac{1}{L_v} \frac{\Delta A_e + \rho_a c_p G_a D}{\Delta + \gamma(1 + G_a/G_i)}, \tag{1}$$

where $L_v$ is latent heat of vaporization (J kg$^{-1}$ K$^{-1}$), $\rho_a$ is the air density (kg m$^{-3}$), $c_p$ is the heat capacity of dry air at constant pressure (J kg$^{-1}$ K$^{-1}$), $D$ is the vapor pressure deficit at air temperature (Pa) and $A_e$ is the available energy (W m$^{-2}$). Depending on specific ET component $E_i$, the surface conductance $G_i$ (m s$^{-1}$) and aerodynamic conductance $G_a$ have different forms. For canopy layer, which contributes to $T_r$ and $E$, the $G_a$ represents efficiency of within-canopy turbulent transport and transport through laminar boundary layers on leaf surfaces, and is computed as a function of wind speed $U$, canopy height $h_c$ and LAI (Magnani et al., 1998; Leuning et al., 2008) (Supp. S2).

### 2.1.2 Transpiration and canopy conductance

To calculate $T_r$ and resulting water uptake from the root zone, an estimate of the canopy conductance $G_c$ is needed. Analysing large corpus of leaf gas-exchange data through stomatal optimization arguments, Medlyn et al. (2012) proposed that leaf-scale stomatal conductance ($g_s$, mol m$^{-2}$ s$^{-1}$) is related to leaf net photosynthetic rate ($A$, $\mu$mol m$^{-2}$s$^{-1}$) as

$$g_s = g_o + 1.6 \left(1 + \frac{g_1}{\sqrt{D}}\right) \frac{A}{C_a}, \tag{2}$$

where $C_a$ is the atmospheric CO$_2$ mixing ratio (ppm), $D$ (kPa) is vapor pressure deficit, $g_o$ residual (or cuticular) conductance and $g_1$ a species-specific parameter that depends on plant water use strategy. Noting that $g_o \ll g_s$ (Medlyn et al., 2012) and representing photosynthetic light response by saturating hyperbola (Saugier and Katerji, 1991), eq. (2) can be approximated as

$$g_s = 1.6 \left(1 + \frac{g_1}{\sqrt{D}}\right) \frac{A_{max}}{C_a} \frac{PAR}{PAR + b} C_{air}, \tag{3}$$

where $A_{max}$ ($\mu$ mol m$^{-2}$ s$^{-1}$) is the light-saturated photosynthesis rate, $b$ (W m$^{-2}$) the half-saturation value of photosynthetically active radiation (PAR), and molar density of air $C_{air}$ (mol m$^{-3}$) converts units of $g_s$ to m s$^{-1}$.

Assuming PAR decays exponentially within the canopy, $PAR(L) = PAR_o exp(-k_p L)$ (where $L$ is the cumulative leaf area from canopy top, $k_p$ the attenuation coefficient and PAR$_o$ the incoming PAR above canopy) and neglecting vertical variations in $D$, eq. (3) can be integrated analytically over $L$ (Saugier and Katerji, 1991) yielding canopy conductance (m s$^{-1}$) as

$$G_c = \left[1.6(1 + \frac{g_1}{\sqrt{D}})\frac{A_{max}}{C_{a,ref}}\right] \left(\frac{1}{k_p} \frac{PAR_o + b}{PAR_o \times exp(-k_p LAI) + b/k_p}\right) C_{air} \times f(\theta_{REW}) \times f_S, \tag{4}$$





where the first term of eq. 4 is the canopy-scale light and $D$ response, the $f(\theta_{REW})$, and $f_S$ (-) are dimensionless scaling factors introduced to represent the effect of soil moisture availability (eq. 6) and phenology. Equation (4) shows that at a given LAI, $G_c$ is constrained leaf by leaf water use traits (via $g_1$), and photosynthetic capacity and light-response (via $A_{max}$ and $b$). Such traits are readily measurable by leaf gas-exchange and widely available in literature and in plant trait databases such as

TRY (Kattge et al., 2011).

Water use strategies and to lesser extent photosynthetic capacity of typical coniferous and deciduous species in boreal forests differ (see e.g. Lin et al. (2015)). Thus, LAI-weighted effective values of $g_1$ and $A_{max}$ are calculated for a grid cell as

$$p = (1 - f_d)\, p_c + f_d p_d \tag{5}$$

where $p$ is the parameter, the subscript $c$ and $d$ refer to conifer and deciduous trees, respectively, and $f_d = LAI_d/(LAI_c +$

$LAI_d)$ the contribution of deciduous trees on total LAI. The seasonal cycle of $LAI_d$ is described using scheme based on accumulated degree-days (Launiainen et al., 2015) calibrated using leaf phenology observations in Southern and Northern Finland.

The effect of soil moisture availability on $G_c$ is adopted from a sap-flow study on Scots pine and Norway Spruce in central Sweden (Lagergren and Lindroth, 2002)

$$f(\theta_{REW}) = \begin{cases} \frac{\theta_{REW}}{x_r}, & \theta_{REW} < x_r \\ 1, & \theta_{REW} \geq x_r, \end{cases} \tag{6}$$

where $x_r$ (-) is a threshold parameter. The relative plant available water $\theta_{REW}$ relates volumetric water content $\theta$ (-) in the root zone to soil hydraulic properties as

$$\theta_{REW} = \frac{\theta - \theta_{wp}}{\theta_{fc} - \theta_{wp}}, \tag{7}$$

where $\theta_{fc}$ and $\theta_{wp}$ (-) soil-type dependent water content at field capacity and wilting point, respectively.

The phenology factor $f_S$ describes seasonal acclimation of photosynthetic capacity as a function of delayed temperature sum (Kolari et al., 2007) and is adopted from Launiainen et al. (2015).

### 2.1.3   Evaporation from forest floor

Forest floor evaporation $E_f$ is extracted from the organic moss/humus layer at the forest floor (Fig. 1). We compute $E_f$ as

$$E_f = f \times E_{f,0}, \tag{8}$$

where $E_{f,0}$ is evaporation rate when moisture supply in the organic layer does not limit $E_f$, and is calculated by eq. 1 where $R_n = R_{n0}\, exp^{-k_p\, LAI}$, $G_a$ depends on surface roughness length and modeled $U$ 0.5m above the forest floor and $G_i$ now represents the conductance of saturated ground surface ($G_f$) and is here calibrated against EC data from a boreal fen as





described later. The factor $f$ accounts for the decay of $E_f$ in drying organic matter as

$$
f = \begin{cases} \frac{\theta_{org}}{x_{r,o}}, & \theta_{org} < x_{r,o} \\ 1, & \theta_{org} \geq x_{r,o}, \end{cases} \tag{9}
$$

$\theta_{org}$ (m$^2$m$^{-2}$) is organic layer water content, and the threshold value $x_{r,o} \sim 0.8\theta_{fc,org}$ based on linear decrease of moss evaporation below a threshold moisture content (Williams and Flanagan, 1996).

### 2.1.4 Interception, throughfall and evaporation from canopy storage

The canopy water storage is described as a single pool filled by interception $I_c$ of precipitation and snowfall, and drained by evaporation/sublimation $E$ and snow unloading $U_s$ (all mm d$^{-1}$). The change in canopy water storage $W$ (mm = kg m$^{-2}$ ground) during model timestep $\Delta t$ is

$$
\frac{\Delta W}{\Delta t} = I_c - E - U_s. \tag{10}
$$

The interception sub-model assumes that full storage is approached asymptotically (Aston, 1979; Hedstrom and Pomeroy, 1998)

$$
I_c = (W_{max} - W_0)\left(1 - e^{-\frac{c_f}{W_{max}}P}\right), \tag{11}
$$

where $c_f$ (-) is canopy closure, $W_0$ the initial water storage, and canopy storage capacity $W_{max} = w_{max}LAI$ (mm) linearly proportional to LAI. The empirical storage parameter $w_{max}$ (mm LAI$^{-1}$) is known to be greater for rain and snow (Koivusalo and Kokkonen, 2002); if $W$ exceeds $W_{max}$ of liquid water and daily mean temperature is above zero, the excess snow storage is removed as snow unloading and added into throughfall input to snow model. In snowfree conditions, all throughfall is routed to forest floor surface and provides input to *Bucket* sub-model.

Evaporation / sublimation from canopy storage is calculated by P-M equation (eq. 1), where the $G_a$ is defined as for $T_r$ while the canopy surface conductance $G_i$ set infinite for evaporation from wet canopy, and computed for snow sublimation as in Essery et al. (2003) and Pomeroy et al. (1998) (Suppl. S4).

### 2.1.5 Snow accumulation and melt

Snowpack on the ground is modelled in terms of snow water equivalent (SWE, mm), a lumped storage receiving throughfall and unloading from the canopy and releasing water by snowmelt. The melt rate $M$ (mm d$^{-1}$) is based on temperature-index approach

$$
M = \min(SWE, K_m(T_a - T_o)), \tag{12}
$$

where $T_o = 0.0°C$ is threshold temperature and $T_a$ daily mean air temperature. The melting coefficient $K_m$ (mm d$^{-1}$ $°C^{-1}$) decreases with increasing canopy closure as Kuusisto (1984)

$$
K_m = K_{m,o} - 1.64c_f, \tag{13}
$$



where $K_{m,o}$ is the melting coefficient at open area. The snowpack can retain only a certain fraction of liquid water (Table 1 and Suppl. S4), and excess is routed to soil module.

## 2.2 Bucket model: topsoil water balance

The topsoil water balance at each grid cell is described as a two-layer bucket model (*Bucket*, Fig. 1). An organic layer of depth $z_{org}$ (mm), representing living mosses and poorly decomposed humus, overlies the root zone and acts as an interception storage for throughfall and snowmelt. Its volumetric water content $\theta_{org}$ (m³m⁻³) is bounded between the field capacity $\theta_{fc,org}$ and residual water content $\theta_{r,org}$ and vary according to

$$\frac{\Delta \theta_{org}}{\Delta t} = \frac{I_{org} - E_f + Q_{r,ex}}{z_{org}}, \tag{14}$$

where $I_{org}$ is the interception rate, restricted either by throughfall or available storage space, and $Q_{r,ex}$ returnflow from the hillslope through the rootzone described below. All $E_f$ is extracted from the organic layer.

The water content $\theta$ in the root zone of depth $z_z$ (mm) changes according to

$$\frac{\Delta \theta}{\Delta t} = \frac{I_f - T_r - D_r + Q_r}{z_s}, \tag{15}$$

where infiltration $I_f$ (mm d⁻¹) and returnflow from the catchment sub-surface storage (sect. 2.3) $Q_r$ provide input, and $T_r$, and drainage $D_r$ outflows from the root zone. The maximum water storage is determined by root zone depth $z_s$ and porosity $\theta_s$, and $I_f$ restricted either by potential infiltration or available storage space. In case of infiltration or returnflow excess, the organic layer storage is first updated, and remaining routed to stream outlet without delay as surface runoff ($Q_s$).

Drainage $D_r$ (mm d⁻¹) from root zone occurs whenever $\theta$ is above field capacity $\theta_{fc}$ as (Campbell, 1985)

$$D_r(\theta) = \begin{cases} K_{sat} \left( \frac{\theta}{\theta_s} \right)^{2\beta+3}, \theta > \theta_{fc}, \\ 0, \theta \leq \theta_{fc}, \end{cases} \tag{16}$$

where the saturated hydraulic conductivity $K_{sat}$ (mm d⁻¹) and its decay parameter $\beta$ depend on soil type.

## 2.3 Topmodel: integration from point to catchment level

To achieve computational efficiency and applicability at large-scale, lateral flow in the saturated zone is not explicitly solved but grid-cell and catchment water balances conceptually linked by Topmodel approach (Beven and Kirkby, 1979). In *Topmodel*, the catchment sub-surface storage is described as a single bucket (Fig. 1). The change in the average saturation deficit $\langle S \rangle$ (mm), i.e. the average amount of water per unit area required to bring the catchment sub-surface storage (below the root zone) to saturation, is

$$\frac{\Delta \langle S \rangle}{\Delta t} = -\langle D_r \rangle + \langle Q_b \rangle + \langle Q_r \rangle, \tag{17}$$

where $\langle D_r \rangle$ (mm d⁻¹) is catchment average root zone drainage, $\langle Q_b \rangle$ (mm d⁻¹) the catchment baseflow and $\langle Q_r \rangle$ (mm d⁻¹) average return flow from the sub-surface storage. Assuming soil transmissivity is spatially uniform and decays exponentially





with depth, the $\langle Q_b \rangle$ becomes (Beven, 1997)

$$\langle Q_b \rangle = Q_o \, exp^{-\langle S \rangle /m} = T_o \, exp^{-\langle TWI \rangle} \, exp^{-\langle S \rangle /m}, \tag{18}$$

where $m$ (mm) is a scaling parameter reflecting the effective water-conducting soil depth, $T_o$ the soil transmissivity at saturation, and $Q_o$ (mm d$^{-1}$) the baseflow rate when $\langle S \rangle$ is zero. The $\langle TWI \rangle$ represents the catchment average of local topographic wetness index $TWI$ defined by the natural logarithm of the area draining through a grid cell $a$ from upslope and tangent of the local surface slope $\beta$ (Beven and Kirkby, 1979)

$$TWI = ln \left( \frac{a}{\tan \beta} \right). \tag{19}$$

The saturation deficit $S$ (mm) of a grid cell is uniquely related to $\langle S \rangle$ by

$$S = \langle S \rangle + m \left( \langle TWI \rangle - TWI \right), \tag{20}$$

which implies that grid cells with high TWI have higher probability to become saturated, and the catchment saturated area fraction is related both to TWI distribution and to the amount of water in the catchment sub-surface storage. Furthermore, eq. (19 shows high value of TWI can result either from large contributing area or flat local topography.

At grid cells where saturation excess ($S < 0$) occurs, returnflow $Q_r = -S$ from the sub-surface storage is routed through the rootzone and organic layer and their water storages are sequentially updated at next $\Delta t$. This creates an approximate feedback from local $S$, controlled by catchment water storage and topography, to topsoil water budget (sect. 2.2) and delays drying of root zone and organic layer at lowland grid cells receiving $Q_r$ from the hillslope.

The specific discharge $Q_f$ (mm d$^{-1}$) at catchment outlet is finally computed as

$$\langle Q_f \rangle = \langle Q_b \rangle + \langle Q_s \rangle, \tag{21}$$

where $\langle Q_s \rangle$ is the catchment average surface runoff (sect. 2.2).

## 2.4 Model inputs

SpaFHy requires daily mean air temperature $T_a$ (°C), global radiation $R_g$ (Wm$^{-2}$), relative humidity RH (%), wind speed (ms$^{-1}$) and daily accumulated precipitation $P$ (mm d$^{-1}$) as forcing. The forcing can be either spatially uniform or vary for each grid cell in the spatial simulations. The available energy is computed from $R_g$ accounting for the effect of LAI on $R_n$ (Fig. 2a in Launiainen et al. (2016)), and PAR$_o = 0.5 \times R_g$.

The model requires following variables to be provided at user-defined grid:

1. Canopy and Bucket -submodels

   – Conifer and deciduous tree 1-sided leaf-area index (LAI$_c$ and LAI$_d$, respectively)

   – canopy height $h_c$ (m)



- Organic layer depth, root zone depth and hydraulic properties (Table S1)

2. Topmodel -submodel

- topographic wetness index TWI

- masks of catchment area and permanent water bodies

All the above variables are derived from open GIS-data available throughout Finland. The SpaFHy structure is modular and the three sub-models are linked via water fluxes, and feedbacks based on state variables such as $\theta_{REW}$ (Fig. 1). Each sub-model can thus be used stand-alone when appropriate forcing data is provided. The model is written in pure Python 2.7/3.6 and uses element-wise operations of Numpy -arrays for all computations. The GIS-data for model initialization are given as raster arrays, while NetCDF -format is used for storing the model outputs that include daily grids of all state variables and fluxes.

**2.5   Model parametrization and sensitivity analysis at stand scale**

Parameters required by each sub-model are given in Table 1 with their generic values. We applied a sequential approach to determine the generic parameter set to describe above-ground hydrology of coniferous-dominated landscape. First, we derived likely ranges of *Canopy* sub-model parameters from the literature and predictions of a common leaf gas-exchange model (Suppl. S3). The rainfall interception capacity is dependent on model timestep $\Delta t$ and calibrated against spatially averaged

throughfall measurements (2001 - 2010) made at the Hyytiälä research station in Juupajoki, Southern Finland (FIHy; Table. 2, Fig. S2). An overview of the site is given by Hari and Kulmala (2005) while Ilvesniemi et al. (2010) describe the hydrological measurements. The surface conductance for evaporation from wet soil surface ($G_f$) was calibrated against eddy-covariance (EC) measured ET from a boreal fen site (FISii, Alekseychik et al. (2017)) located next to FIHy (Table 2). Monte-Carlo simulations (n=100), where parameter candidates were sampled from uniform distribution and objective function was set to

minimize bias between modelled and measured values, were performed.

   After parameter ranges were determined, a global sensitivity analysis was performed to identify the key parameters controlling annual ET and its components, and annual maximun SWE. For this analysis, the *Canopy* and *Bucket* modules were coupled and the resulting stand-scale model run with various parameter combinations using daily forcing data from FIHy (2000-2010) as input. We used Morris method, a global extension of elementary effect test used to determine which model parameters are

negligible, linear and additive, or non-linear or involved in interactions with other parameters (Morris, 1991; Campolongo et al., 2007). In Morris method, three sensitivity measures are calculated from the distribution of scaled elementary effects. The mean of distribution ($\mu$) is the overall effect of a parameter on the output, and the standard deviation ($\sigma$) is effect of a parameter due to non-linearity or due to interactions with other parameters. Third measure is the mean of the distribution of the absolute values of the elementary effects ($\mu^\star$) (Campolongo et al., 2007). The $\mu^\star$ provides ranking of parameters which is not

biased by possible non-monotonic behavior of the model. The sensitivity measures are interpreted graphically together with rank parameters according to their overall influence on outputs. To ease graphical interpretation, standard error of the mean as $SEM = \sigma/\sqrt{r}$, where $r$ is the number of trajectories, was estimated and used as suggested by Morris (1991). Analysis was conducted by using a Python package SALib (v. 1.1.2; Herman and Usher, 2017).



The ranges of the 14 parameters considered in the sensitivity analysis are listed in Table 3. In the analysis, leaf area indices for conifers and deciduous were calculated from total 1-sided LAI and deciduous fraction. Each parameter was allowed to vary over eight levels, and 60 optimal trajectories were generated from 600 initial trajectories by the sampling scheme introduced by Ruano et al. (2012). In total, 900 samples were generated and the number of optimal trajectories was determined following

Ruano et al. (2011).

After sensitivity analysis, most of the parameters could be fixed (those deemed less-influential), and only the 'generic' values for $A_{max}$ and $g_1$ in eq. (4) were confirmed by calibrating them against eddy-covariance (EC) -measured ET (years 2005 - 2007) at FIHy -site. The possible ranges of these parameters were constrained by physiological arguments. Monte-Carlo simulations (N=100), where parameters were sampled from uniform distribution and objective function was set to minimize bias between

modelled and measured daily ET were performed. We considered only dry-canopy conditions, i.e. no rain during the current or previous day.

### 2.6 Model validation at stand and catchment scales

#### 2.6.1 Stand scale

To validate how daily ET can be predicted across LAI, sitetype and latitudinal gradient using a single parameter set (Table 1),

the stand-level model was run using daily meteorological data from nine additional EC-flux sites in Finland and Sweden (Table 2, Fig. S1). The sites range from dense mixed coniferous forests (SENor) to recently harvested stand (FICage4) and pristine fen peatland site (FISii), and the measurements, flux calculation and data post-processing have been described elsewhere (Launiainen et al., 2016; Minunno et al., 2016). For each site, $LAI_c$, $LAI_d$, $h_c$ and soil properties were set according to measured/inferred values, and predicted daily growing-season (May-Oct) ET in dry-canopy conditions ($ET \simeq T_r + E_f$) was

compared to measured. At FIHy, the soil moisture in the root zone was measured continuously, and SWE recorded bi-weekly during five winters and used to compare respective model predictions.

#### 2.6.2 Catchment scale

To address how well SpaFHy can predict daily specific discharge and annual partitioning of $P$ into ET and $Q_f$ at catchment scale, we applied the model to 21 small boreal headwater catchments located throughout Finland (Fig. S1, Table S2) using same

generic parameter set in the stand-level validation (Table 1). All the catchments belong to the Finnish network for monitoring water quality impacts of forestry (Finér et al., 2017), and their characteristics can be found in Supplementary material (Table S2). The water levels at v-notch weirs were measured continuously at the catchment outlets by limnigraphs or pressure-sensors, and manual reference measurements ware taken ca. 20 times per year adjacent to water quality sampling and used to calibrate the weir water level data whenever necessary. Weir equations and catchment area were used to convert water level to specific

discharge $Q_f$. In absence of *in-situ* weather data, daily 10 x 10 km grid data provided by Finnish Meteorological Institute were used as model forcing taking the values from a gridpoint nearest to the catchment outlets. Since wind speed was not available, it was set to constant value of 2 ms$^{-1}$ resembling annual mean 2 m wind speed in Finland.





### 2.6.3 Processing of GIS -data

Example of GIS-data used to set up the model for catchment C3 in Eastern Finland are shown in Fig. 2. The catchment boundaries and TWI were derived from DEM provided by National Survey of Finland (NSLF, 2017). The DEM original resolution was 2 m or 10 m depending on catchment location. The resolution was aggregated with the mean elevation value

into 16 m resolution which corresponds to the resolution of the multi-source National Forest Inventory of Finland (mNFI) data. The mNFI data provides the essential data layers for the model, e.g. stand volume, basal area, mean height, age, site fertility class and estimates of root, stem, branch and needle/leaf biomasses for pine, spruce and aggregated deciduous trees at 16 m resolution throughout Finland Mäkisara et al. (2016) and thus 16 m was chosen as the baseline resolution.

The DEM pre-processing, defining of the catchment boundaries and the calculation of TWI based on the aggregated DEM

were conducted in WhiteBox GIS programme (Lindsay, 2014). Pre-processing included consideration of the road and stream intersections derived from the Topographical Database (NSLF, 2017), which were burned into the DEM to account for culverts and ensure continuous stream network. Further, all water elements were burned into the DEM with 1 meter upper threshold and a decay factor accounting for possible miss-aligned stream data. The filling of artificial pits in DEM was conducted using 'Fast Breach Depressions' tool (Lindsay, 2016) and the flow direction and flow accumulation ($a$) rasters were calculated with

the D-infinity method (Tarboton, 1997). The TWI was finally calculated by eq. (19) and small lakes within the catchments, derived from the Topographic Database (NSLF, 2017), were reset as nodata and omitted from further computations. The needle and leaf mass rasters were converted into $LAI_c$ and maximum deciduous tree LAI $LAI_{d,max}$ using specific 1-sided leaf-areas for pine, spruce and birch (6.8, 4.7 and 12.0 m$^2$ kg$^{-1}$, respectively; Härkönen et al. (2015)). The canopy closure and $h_c$ were obtained directly from the mNFI data.

Topsoil classification was derived from soil maps and peatland boundaries. Soil information is provided for parts of Finland in 1:20 000 scale while the whole Finland is covered with a coarser 1:200 000 scale soil map (GSF, 2015). Peatland classification is available as detailed polygon elements from the Topographical Database. The soil information were transformed to the 16 m grid based on the majority principle, and then re-classified into four classes: coarse, medium and fine-textured mineral and organic peat soils whose hydrologic properties are given in Table S1. Fine-textured soils correspond to clayey and silt

soils, whereas coarse-textured are fine sand and coarser. Majority of the mineral soils in the study catchments belong to the medium-textured class (Table S2).

### 2.6.4 Calibration of Topmodel against measured specific discharge

Catchment-specific calibration was performed to determine the effective soil depth $m$ of *Topmodel*, a parameter that defines the shape of $Q_f$ recession and catchment average storage deficit $\langle S \rangle$ (eq. 20). The parameter $T_o$ was fixed to 0.001 ms$^{-1}$ since

it was found not markedly affect the model performance, as also observed elsewhere (Beven, 1997). The $m$ was calibrated against measured daily specific discharge using Monte-Carlo sampling from uniform distribution (N=100). We used modified



Willmott's index of agreement (Krause et al., 2005) as an objective function to quantify the goodness of fit

$$d_j = 1 - \frac{\sum_{i=1}^{n} |\langle Q_{m,i} \rangle - \langle Q_{f,i} \rangle|}{\sum_{i=1}^{n} |\langle Q_{m,i} \rangle - \langle \overline{Q_m} \rangle|}, \tag{22}$$

where $d$ is in a range of 0 to 1, the higher the value, the better the match is; $\langle Q_{f,i} \rangle$ and $\langle Q_{m,i} \rangle$ are modelled and measured specific discharges at day $i$, and $\overline{\langle Q_m \rangle}$ represents temporal average of the measurements. This model goodness statistics provided
visually determined better fits of streamflow recession than other commonly used statistical criteria, e.g. Nash-Suchliffe model efficiency that was overly sensitive to high-flow peaks and affected by potential biases in $P$. The initial state of the model was set through one year spin-up period. The value of $m$ significantly affects the dynamics of specific discharge $\langle Q_f(t) \rangle$ and $\langle S(t) \rangle$ but had negligible impact on catchment $\langle \overline{ET} \rangle$ or $\langle \overline{Q_m} \rangle$ at annual scale.

## 3 Results

### 3.1 Sensitivity analysis at stand scale

The sensitivity measures $\mu$ and $\sigma$ for maximum SWE and annual ET and its components are shown in Table 4, and the ranking of parameters (via $\mu_\star$) in Supplementary Fig. S2.

Total LAI was ranked the most influential parameter for all studied *Canopy* sub-model outputs. In addition to $LAI$, the parameters that affect leaf level water use ($g_1$, $z_s$, $A_{max}$, and $b$) were among the most influential parameters for total ET and transpiration. The most influential parameters for ground evaporation $E_f$ were LAI and $k_p$, which define radiation availability at the ground layer. LAI also affects wind speed and thus aerodynamic conductance at the ground layer. In addition, surface conductance for wet forest floor $G_f$ and $z_{s,org}$ and $\theta_{fc,org}$ that define water storage capacity of the organic layer were significant for $E_f$. The most influential parameters for interception evaporation ($E$) were LAI, $w_{max}$, $f_d$, $h_c$, and $w_{max,snow}$ that define interception capacity and subsequent evaporation/sublimation of rain and snow. The most influential parameters affecting annual maximum snow water equivalent (SWE) were LAI, $w_{max,snow}$, $f_d$, $w_{max}$, and $h_c$.

LAI had also the largest $\sigma$ meaning either interactions with other parameters or strong non-linearity. In case of ET and $T_r$, coefficient of variation ($\sigma$ / ($\mu^\star$ – ratio) was over 1.0 and for E, $E_f$, and SWE it was smaller but over 0.5. The most influential parameters of all studied outputs had the coefficient of variation over 0.5. Non-monotonic behavior (i.e. $\mu$ / $\mu_\star$ –ratio is significantly different from unity) of the model was only observed in case of ET for LAI.

### 3.2 Validation at stand-scale

The predicted daily dry-canopy ET and root zone moisture content are compared against 10 years of measurements at FIHy -site in Fig. 3. The results indicate the model reproduces well the observed seasonal patterns of ET and $\theta$ both during calibration (2005 - 2007) and validation period. The regression plots indicate ET predictions have negligible bias and well represent the variability while the soil moisture changes are not fully captured. The SWE (Fig. 10) was also well reproduced by snow model, even parametrized by literature values (Pomeroy et al., 1998; Essery et al., 2003).





The ET predictions for the nine additional EC-sites are shown in Fig. 4. The growing season (doy 120 - 273) dry-canopy ET is reasonably well predicted compared to independent observations across broad LAI -range (from 0.7 to 6.8 $m^2m^{-2}$) and over latitudinal and site-type gradient (Table 2, Fig. S1). At the youngest, recently clearcut site FICage4 the model underestimates ET, while slight overestimation is observed in particular at the northernmost, old-growth Scots pine site on coarse textured

soil (FISod). In terms of explained variability, the model performance is the weakest at SESky2, FIKal and FILet, potentially because ill-represented moisture limitations of transpiration and/or that of $E_f$. The non-linear behavior at SENor and less clearly at SESky2 and FILet is primarily caused by slower than observed spring recovery at the sites having high abundance of Norway spruce (not shown). As the Norway spruce has observed to recover more rapidly from winter dormancy than pine (Linkosalo et al., 2014; Minunno et al., 2016), this can be partly related to biased phenology-model that is based on Scots pine

(Kolari et al., 2007).

Also ET at the pristine fen peatland site FISii, where $E_f \gg T_r$, was accurately predicted when moisture limitation of $E_f$ was neglected ($f = 1$ in eq. 8). Such case can be expected due to strong capillary connection between peat moss (*Sphagnum sp.*) and shallow water table maintained by lateral inflows from the surrounding landscape and weak drainage (Rouse, 1998; Ferone and Devito, 2004). When the organic layer moisture content feedback to $E_f$ was activated, however, the ET at FISii

was frequently underestimated during summer dry spells (not shown); in the point-scale simulations this represent the case where the organic layer water storage is recharged only by $P$.

Overall, the model performance at stand scale was satisfactory and dry-canopy ET well predicted over range of forest sites and climatic gradient in Finland. This suggests that the proposed three-source ET formulation and its generic parametrization for $T_r$, $E_f$ and snow interception should be scalable over landscape scale variability of LAI, site types and latitude-driven

weather forcing. Since EC measurements are know to be problematic during rainfall events (van Dijk et al., 2015; Kang et al., 2018), the comparison of stand-level ET was restricted to dry-canopy conditions.

### 3.3    Catchment water balance and specific discharge

On annual scale, changes in catchment water storage are negligible compared to annual $\langle ET \rangle$ and $\langle Q_f \rangle$, and water balance approach provides an independent check for the upscaled ET predictions at catchment level. Fig. 5 shows the comparison of

modelled and water-balance based annual evapotranspiration fraction $\left\langle \overline{ET/P} \right\rangle$ for the 21 headwater catchments across Finland (Fig. S1, Table S2). Results show a close agreement between measured and modelled $P$ partitioning across the catchment space, especially considering the uncertainties in both axis.

The uncertainty range of modelled $\left\langle \overline{ET/P} \right\rangle$ implies the impact of model parameter uncertainty. The uncertainty range in Fig. 5 was derived by varying the most influential parameters for total ET and its partitioning (LAI, $g_1$, $w_{max}$, $w_{max,snow}$) by

$\pm\,20\,\%$ and grouping the combinations into 'high' and 'low' ET scenarios, respectively. While the model is mass-conserving, uncertainty of $\left\langle \overline{ET/P} \right\rangle$ derived from catchment water balance is linearly proportional to uncertainty of $Q_f$ derived from streamflow measurements and catchment area. Also systematic and random errors in the annual $P$ cause respective uncertainties in $\left\langle \overline{ET/P} \right\rangle$. In Fig. 5 the horizontal errorbars correspond to modest 10 % uncertainty assumed for $P$ and catchment area.





Overall, the model predictions are reasonably good across the catchment space. Stepwise linear regression was tested to explain the annual residuals by catchment characteristics in Table S2 but no significant relationships were found. Also inter-annual variability of $\langle \overline{ET/P} \rangle$ was well captured for majority of the catchments (not shown).

Figure 6 shows specific discharge at catchment C3 in Eastern Finland over two years characterized by wet (2012, P 452 mm in June-Sept) and dry (2013, P 246 mm) growing seasons. In 2012, the high snow accumulation resulted into stronger streamflow peak, and frequent rainfall events kept the catchment average root zone moisture $\langle \theta \rangle \geq 0.3$ m$^2$m$^{-2}$ throughout the year (Fig. 7). Also $Q_f$ remained significantly higher throughout the summer compared to 2013, and responded rapidly to rainfall. During the drier 2013, transpiration depleted the root zone moisture well below field capacity and $\langle \theta \rangle$ dropped frequently to $\sim$0.15 m$^2$m$^{-2}$ in June - August. The model was well able to predict the spring $Q_f$ peaks and recession curve, and also rainfall-induced peaks during the wet summer. During drier conditions, however, the small-magnitude peaks in summer $Q_f$ were not well captured by the model. This suggests too high *Bucket* storage capacity and thus underestimated fraction of saturated area that contributes to overland flow during and after precipitation events. This is, however, not a general behaviour of the model as better comparison between measured and modelled specific discharge was observed at several other catchments (not shown).

## 3.4 Within-catchment variability

### 3.4.1 Soil moisture

To illustrate how vegetation, soil and topography create within-catchment variability to local water fluxes and state variables, we use a small 70 ha catchment C3 (Porkkavaara) located in Eastern Finland (Table S2) as an example. The average root zone moisture $\langle \theta \rangle$ and its spatial standard deviation $\sigma_\theta$, vary strongly over the hydrologic year, as shown for two contrasting years (Fig. 7). Snapshots of spatial variability of $\theta$ and local saturation deficit of Topmodel (eq. (20)) are shown for dry and wet conditions in Fig. 8.

The primary fluxes and landscape factors driving modelled spatial variability of soil moisture depend on antecedent soil moisture conditions (Fig. 7). During winter, root zone moisture content decreases and its spatial variability is dampened by slow drainage. The onset of snowmelt is followed by infiltration peak and saturated soils nearly throughout the catchment (Fig. 8). This leads to rapid increase of $\sigma_\theta$, mainly because of spatial variation in soil porosity. In wet 2012, drainage rapidly decreased $\sigma_\theta$ after snow melt, while the spatial variability was preserved in the drier 2013 until early July. The latter is related to spatially heterogeneous transpiration rate (Fig. 9) that create spatial variance of soil moisture and compensate the dampening effect of drainage until ca. doy 180. After that $\sigma_\theta$ starts to decrease because transpiration at grid cells characterized by coarse and medium-textured soil and high LAI (Fig. 2) become soil-moisture limited (eq. 7). In 2013 summer when $\langle \theta \rangle$ was most of the time well below field capacity, the rainfall events tend to dampen spatial variability of soil moisture (Fig. 7). In wetter conditions (most of 2012, autumn 2013), however, the effect of infiltration is opposite and resembles that of spring snowmelt.

As a result, there is clear hysteresis of $\sigma_\theta$ with respect to antecedent $\langle \theta \rangle$ in the dry year while such patterns are less visible in moist conditions. This indicates soil and vegetation variability can in the model both create or destroy spatial variability of soil



moisture, as has been proposed both by theoretical arguments (Albertson and Montaldo, 2003) and analysis of soil moisture observations (Teuling and Troch, 2005). Moreover, during drier spells the spatial soil moisture variability was primarily driven by heterogeneous vegetation and plant water use, while soil type and topography are the primary controls in wet conditions and outside growing season (Seyfried and Wilcox, 1995; Teuling and Troch, 2005; Robinson et al., 2008).

### 3.4.2 ET and snow

The predicted spatial variability of evaporation fraction $\overline{ET/P}$ and its components are illustrated in Fig. 9. The model results, averaged over 2006 - 2016 period, reveal strong sensitivity of component ET fluxes to stand leaf area index, and secondary impacts of soil type and topography. The model predicts $\overline{ET/P}$ increases non-linearly with LAI and varies from $>0.25$ at grid cells where LAI $< 1$ $\mathrm{m^2 m^{-2}}$ to $\sim 0.65$ at locations where the standing tree volume and LAI (Fig. 2) are largest. The shape of LAI-response results from the non-linear scaling of component fluxes with LAI, which also explain the inflection point at LAI $\sim 3$ $\mathrm{m^2 m^{-2}}$.

Interception of rainfall and snow contributes from less than 5 to 30 % of long-term $P$, which is in line with measurements from boreal forests (Barbier et al., 2009; Toba and Ohta, 2005). The linear scaling of interception capacity with LAI and asymptotic approach of full storage (eq. 11), and temporal distribution of precipitation lead to the near-linear increase of $\overline{E/P}$ with increasing LAI (Fig. 9). At grid cells with high fraction of deciduous trees, low wintertime LAI leads to weaker snow interception and smaller annual $\overline{ET/P}$ compared to coniferous-dominated stands.

The spatial patterns of maximum SWE (Fig. 10 indicate snow accumulation in the densest stands (LAI $> 7$ $\mathrm{m^2 m^{-2}}$) was $\sim 75$ % of that on open areas; the exact fraction was found sensitive to winter weather conditions being lowest during mild winters in the southern catchments, and in years with smaller annual snowfall (not shown). The predicted impact of forest canopy on snow accumulation is in good agreement with observational studies from similar climatic conditions in Finland and Sweden (Koivusalo and Kokkonen, 2002; Lundberg and Koivusalo, 2003), although also higher snow interception losses have been reported especially in coastal climates (see Kozii et al. (2017) for summary). The near linear increase of snow interception and resulting decrease of SWE is supported by Hedstrom and Pomeroy (1998); Pomeroy et al. (2002).

The model predictions suggest transpiration contributes from $< 10$ to more than 35 % of annual $P$ being the largest ET component in stands whose LAI $> 1.5$ $\mathrm{m^2 m^{-2}}$ (Fig. 9). The shape of $T_r$ to LAI -response is to most extent caused by saturation of $G_c$ because of light limitations in dense stands (eq. 4). The more liberal water use strategy of deciduous species ($g_{1,d} > g_{1,c}$, Table 1) is reflected as higher transpiration rate at grid cells where deciduous trees form a significant part of total LAI. Moreover, the lower envelope of the points occur at grid cells corresponding to coarse-textured soils (Fig. 2), where drought limitations become most frequent. This is particularly visible also in $\overline{ET/P}$ at LAI $> 2$ $\mathrm{m^2 m^{-2}}$.

Evaporation from forest floor $\overline{ET/P}$ decreases asymptotically with LAI, showing complementary relationship to $T_r$, as expected by decreased available energy in denser stands. The upper envelope curve corresponds to grid cells with high TWI and large tendency to be permanently saturated due returnflow from the hillslope. In these grid cells $E_f$ is mainly determined by available energy; however, rapid drying of the forest floor in sparse stands between rainfall events decreases $\overline{E_f/P}$ and explain it less steep decrease with LAI at grid cells receiving less frequent or no returnflow (lower TWI).



## 4 Discussion

### 4.1 Model capabilities and limitations

At stand-scale, SpaFHy was shown to well reproduce daily ET measured by eddy-covariance technique at several forest and peatland sites in Finland and Sweden (Fig. 3 and 4). The good performance using generic parametrization derived mainly from literature sources and process-specific data suggests the model is capable of accounting for the key drivers of temporal and site-to-site variability of ET. The sensitivity analysis reveals that for given meteorological forcing, total LAI is the primary parameter affecting ET and its partitioning into component fluxes. In case of transpiration rate, the dominant role of LAI and parameters defining leaf water use efficiency ($g_1$ and $A_{max}$) and insensitivity to parameters related to aerodynamic conductance of the P-M equation (eq. 1) indicate variations in $T_r$ are mainly governed by that of canopy conductance (eq. 4). The root zone depth, soil hydraulic properties and size of interception storage in the organic layer ($z_{org}$ and $\theta_{fc,org}$) are important for probability of drought occurrence and consequent reduction of transpiration (Table 4, Fig. 9). Since the root zone depth and soil properties cannot accurately be determined from available GIS-data, these are also parameters where subjective choices and uncertainty may have greatest impact on the results.

The multiplicative formulation for canopy conductance (eq. 4) was derived by coupling the commonly used unified stomatal model (Medlyn et al., 2012) and leaf-scale light response with simplified canopy radiative transfer scheme (see Suppl. S3). The approach accounts the non-linear scaling between $G_c$ and $g_s$ similarly as Saugier and Katerji (1991) and Kelliher et al. (1995). To derive bulk surface conductance for remote-sensing applications, Leuning et al. (2008) combined their $G_c$ scheme with a ground evaporation model based on equilibrium evaporation (Priestley and Taylor, 1972). They showed that after site-specific optimization, the dry-canopy ET was accurately predicted by P-M equation across different vegetation types. That particular model, however, still requires an arbitrary and non-measurable maximum $g_s$ and few other parameters to be specified or calibrated. In our work $g_s$ and its response to $D$ were derived from stomatal optimization arguments and are tightly constrained by plant water use traits and photosynthetic capacity. These traits start to be widely available in databases such as TRY (Kattge et al., 2011), and can also be readily measured using leaf gas-exchange techniques. Due these constraints to $g_s$, we consider eq. (4) as a major advancement of the Leuning et al. (2008) approach. The good comparison between modelled and measured dry-canopy ET for sites having strongly different $T_r/ET$ and $E_f/ET$ -ratios (Fig. 3 and 4) are indeed supportive for the proposed $G_c$ formulation. However the comparison was done within a single vegetation type and further evaluation across ecosystem types are necessary to extend the approach outside boreal forests.

The sensitivity analysis (Table 4 and Fig. S2) proposes the P-M equation could be replaced with simpler approaches. Making the assumption that canopy is well-coupled to the atmosphere, reasonable for aerodynamically rough boreal forests, leads to $T_r = G_c \frac{D}{p_a}$, where $p_a$ (kPa) is the ambient pressure. Also evaporation from the ground and canopy storage were found relatively insensitive to aerodynamic terms, which suggests they could be computed proportional to equilibrium evaporation $E_i = \frac{\alpha_i}{L_v} \frac{\Delta R_{n,i}}{\Delta + \gamma}$, where $\alpha_i$ is a proportionality factor calibrated against measurements, and $R_{n,i}$ available energy at the ground level. Moving to such approaches would relax input data requirements by eliminating the canopy height and wind speed from model forcing.



Open GIS data on LAI, species composition, soil type and topography was used to apply SpaFHy at 16 x 16 m grid size to 21 headwater catchments in Finland. Results indicate the model well reproduces the variability of annual evaporation fraction across catchments (Fig. 5), as well as inter-annual variability at most of the studied catchments (not shown). It should be noted, however, that the variability of annual $\left\langle \overline{ET/P} \right\rangle$ across the catchment space is dominated by latitudinal climate gradient and further testing across different catchments on similar climatic conditions is needed.

The results of site and catchment scale validation suggest that ET and water budget partitioning in boreal forest-dominated landscape can be reasonably well predicted by the model based on generic parametrization, which is advantageous for scalability and applicability of the model for areas and locations where data is scarce or lack for model calibration. Moreover, the model-data comparison at catchment scale supports our initial hypothesis that the ET components and water budgets can be upscaled from stand to catchment scale using relatively simple mechanistic approach that derives characteristics of the modelling domain from open GIS data.

In SpaFHy, the above-ground hydrology and root zone water balance (eq. 14 & 15) are solved distributively (Fig. 1), which propagates the spatial variability of vegetation (LAI, $c_f$, species composition) and soil type into the local hydrological fluxes, SWE and organic layer and root zone water contents. Applied stand-alone, such approach would assume grid cell water balances are independent from each other, and omits the lateral flows and the topographic position of a grid cell on a hillslope. The role of *Topmodel* (sect. 2.3) can be considered as a non-linear streamflow generation routine, which delays average root zone drainage signal $\overline{D_r}$ leading to realistic response of streamflow to $P$ as controlled by TWI distribution. The other catchment properties are lumped into the parameter $m$, the effective subsurface water-conducting depth. It is this parameter that primarily controls both the shape of rainfall-runoff response and streamflow recession. The SpaFHy can thus be used as a simple catchment model to predict the signals of vegetation changes, forest management or varying climatic drivers on streamflow at daily or longer time scales. Indeed, the daily time series of streamflow (Fig. 6) were well reproduced for majority of the 21 studied catchments although $m$ was the only parameter specifically calibrated for each catchment (Suppl. Table 2).

On the other hand, SpaFHy can assist in mapping how soil saturation may vary spatially and temporally as response to weather forcing (Fig. 8). The TWI-based scaling in *Topmodel* is used to predict magnitude and location of returnflow formation based on state of the catchment sub-surface storage. The spatial $Q_r$ field is then used to update *Bucket* sub-model water storages and $\theta$ at respective grid-cells. In this way, SpaFHy can be used to predict local soil saturation that depends on both local (via vegetation and soil characteristics) and approximative landscape (via topography) controls (Fig. 8). In essence, the effect of returnflow formation is to delay drying of gridcells that receive water from the surrounding landscape. Depending on TWI-distribution and value of $m$, this conceptualization implies that some gridcells never receive water from the surrounding landscape(those with low TWI) while some receive $Q_r$ in highflow conditions but not in baseflow conditions. At the other extreme, there are permanently inundated areas (high TWI) that contribute constantly to overland flow. We emphasize that linking grid-cell water balances through *Topmodel* is conceptual rather than physically correct approach (Beven, 1997; Seibert et al., 1997; Kirkby, 1997), and driven by the goal to develop a simple and practically applicable representation of topographic controls of soil moisture. Future work should explore whether $m$ can be related to catchment characteristics to derive a more generic parametrization for *Topmodel*, as well as analyse the impact of parameter uncertainty on streamflow and saturated area




predictions. For applications requiring more rigorous treatment of sub-surface flows, the *Topmodel* can be replaced with 2D ground water flow schemes.

Fig. 8 and 9 show that landscape position (accounted via TWI) can markedly affect gridcell ET and soil moisture. In this work, other topographic controls were omitted for simplicity. While likely to have small impact for annual catchment water
balance, including topographic effects on radiation (Dubayah and Rich, 1995) is presumed to alter the spatial patterns of ET and $\theta$. In addition, the shading by vegetation at the neighbouring grid cells should be considered to derive a more comprehensive understanding on hydrological variability on the landscape. Also adding sub-models to simulate spatial and temporal patterns of soil temperature and frost depth, vegetation productivity and carbon balance would be relatively straightforward future developments.

Validation of spatial predictions of $\theta$, ET or SWE (Fig. 8 - 10) was not attempted in this work. This would require either extensive spatially distributed and continuous *in situ* measurements, or high-resolution (i.e. order of tens of meters) remote sensing data that can be already obtained by near-ground microwave radiometry or low-frequency radars using unmanned aerial vehicles as a platform (Robinson et al., 2008). Also ongoing advances in satellite-based soil moisture (Chen et al., 2014) and ET products (Hu et al., 2015) could be used to evaluate the modelled spatial patterns and temporal evolution of these
hydrological components. A direct validation may, however, be unrealistic since predictions of $\theta$ are inherently dependent both on model formulation, and accuracy of input data such as soil textural information, hydraulic properties and depth of the root zone. The same concerns ET patterns that are affected by uncertainties of LAI and other vegetation characteristics, and degree of complexity the impacts of spatially variable radiation and sub-surface hydrology are accounted for.

As shown in this work, the mNFI data (Mäkisara et al., 2016) can provide estimates of LAI, canopy height, site type and
conifer/deciduous composition at 16 x 16m resolution throughout Finland. Härkönen et al. (2015) compared mNFI-based LAI estimates against ground-based estimates and MODIS LAI, and found good agreement between the methods. Consequently, the mNFI data can provide an easy way to obtain vegetation characteristics for hydrological and biogeochemical models at spatial scale currently unresolved by e.g. MODIS and other satellite products. Similar high-resolution data on forest resources is openly available also from other Nordic countries (Kangas et al., 2018).

### 4.1.1 Potential applications

This study presented a semi-distributed model for boreal forest hydrology at stand and catchment scales (Fig. 1). The model consists of three independent components: a *Canopy* model for above-ground hydrology, a *Bucket* model for topsoil water balance and *Topmodel* for point to catchment integration. The modularity of SpaFHy provides clear advantages since all model components are independently parametrized which allows their stand-alone development and use, as well as inclusion to other
distributed or lumped hydrological models. Moreover, parameters of each sub-model were obtained separately and calibrated based on good-quality data that clearly enhances the predictive power of SpaFHy by reasonably constraining the degree of freedom in model parametrization (Jakeman et al., 2006; Jackson-Blake et al., 2015).

As high resolution GIS data including topographical, soil and forest attributes starts to be increasingly available across boreal region, the proposed approach can provide supports to a variety of questions benefiting from spatial and temporal





hydrological predictions. These include, but are not limited to: (1) predicting soil moisture necessary for e.g. forecasting forest soil trafficability (Vega-Nieva et al., 2009; Jones and Arp, 2017), precision forestry, and confronting climate-induced risks (Muukkonen et al., 2015); (2) identifying how saturated areas, considered as biogeochemical and biodiversity hotspots particularly sensitive to negative environmental impacts of human activities, evolve in time (Laudon et al., 2016; Ågren et al.,

2015); (3) addressing the impacts of forest structure, management and climate change on ET partitioning, streamflow dynamics and soil moisture (Zhang et al., 2017; Karlsen et al., 2016); (4) supporting water-quality modelling in headwater catchments (Guan et al., 2018); and (5) providing starting point for developing spatially distributed forest productivity and sustainability framework that combines open data streams, statistical approaches and mechanistic models. Moreover, we propose the *Canopy* sub-model, in particular the leaf-to-canopy upscaling of canopy conductance, to be tested more widely in boreal and temperate

forest ecosystems.

## 5   Conclusions

A spatially distributed hydrological model for upscaling ET and other hydrological processes from a grid cell to a catchment level using open GIS-data and daily meteorological data was presented and validated for boreal coniferous-dominated forests. SpaFHy consists of three coupled stand-alone modules for aboveground, topsoil and subsurface domains, respectively. An

improved approach to upscale stomatal conductance to canopy scale was proposed, and a generic parametrization of vegetation and snow-related hydrological processes for Nordic boreal forests derived based on literature and data from a boreal FluxNet site. With the generic parametrization, SpaFHy was shown to well reproduce daily ET across conifer-dominated forest stands whose LAI ranged from 0.2 to 6.8 $m^2 m^{-2}$. Predictions of annual ET were successful for the considered 21 boreal headwater catchments in Finland located from 60 to 68 °N, and daily specific discharge could be reasonably well predicted for majority

of the catchments by calibrating only one parameter against streamflow data. In subsequent studies, the model will be used to support forest trafficability forecasting, and predicting the impacts of climate change and forest management on site and catchment water balance.

*Code and data availability.*  The SpaFHy source code (Python 2.7/3.6), a brief user manual and a sample dataset to run the model for a single forest stand and for a single catchment are available under CC BY 4.0 license at www.github.com/lukeecomod/spafhy_v1. Data

from a Hyytiälä (FIHy) used in stand-scale evaluation is available at https://avaa.tdata.fi/web/avaa/-/smartsmear. Eddy-covariance data from other sites can be obtained from the corresponding author. The specific discharge data used in Topmodel calibration and catchment scale evaluation can be obtained from the corresponding author. All GIS-data used in this work is openly available for whole Finland; the entry-point for obtaining GIS-data in Finland is https://www.paikkatietoikkuna.fi/?lang=en. The mNFI -data at 16 m resolution is available at http://kartta.luke.fi/index-en.html.

*Competing interests.*  The authors declare that they have no conflict of interest.



*Acknowledgements.* This study was financially supported through the following projects: the Academy of Finland CLIMOSS (no. 296116 & 307192) and FOTETRAF (no. 295337); Swedish Research Council for Environment, Agricultural Sciences and Spatial Planning (FORMAS, grant no. 2018-01820) and EU Life+ FRESHHABIT. We acknowledge University of Lund (sites SENor, SEKno, SESky2) and University of Helsinki (FIHy, FICage, FISii) for providing the eddy-covariance data for model validation. The Finnish Meteorological Institute is acknowledged for providing the eddy-covariance data (FISod, FIKal, FILet) and 10x10km weather data. The catchment monitoring network used in this work is funded by the Ministry of Agriculture and Forestry of Finland.





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





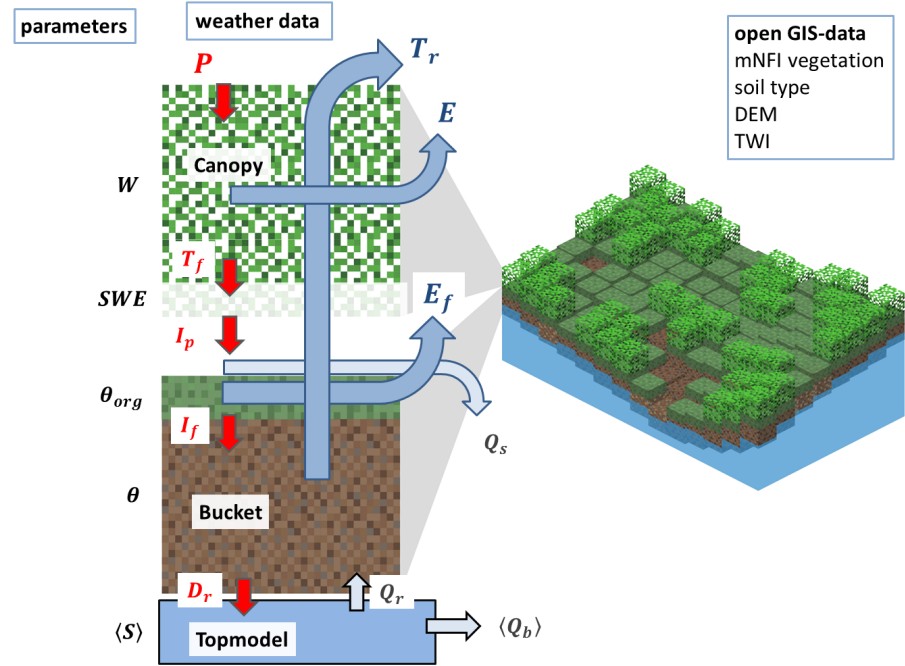

**Figure 1.** Structure of SpaFHy. At each grid cell, above ground and topsoil hydrology is solved by *Canopy* and *Bucket* sub-models whereas lumped *Topmodel* is used to model saturated zone. The arrows correspond to interfacial fluxes: $P$ precipitation; $T_f$ throughfall to virtual snowpack; $I_p$ potential infiltration to organic layer; $I-f$ infiltration to root zone; $D_r$ drainage to saturated zone; $E$ evaporation/sublimation of canopy storage; $E_f$ evaporation from ground; $T_r$ transpiration; $Q_r$ returnflow; $Q_s$ surface runoff; $\langle Q_b \rangle$ baseflow.





**Figure 2.** Spatial data at 16 m resolution used to set up the model for the catchment C3, Porkkavaara, in Eastern Finland (see Table S2). LAI is total 1-sided leaf-area index; $f_d$ deciduous fraction; $h_c$ canopy height; DEM elevation; TWI topographic wetness index; soiltype refers to Table 2. Rasters overlay topographic basemap provided by National Survey of Finland.





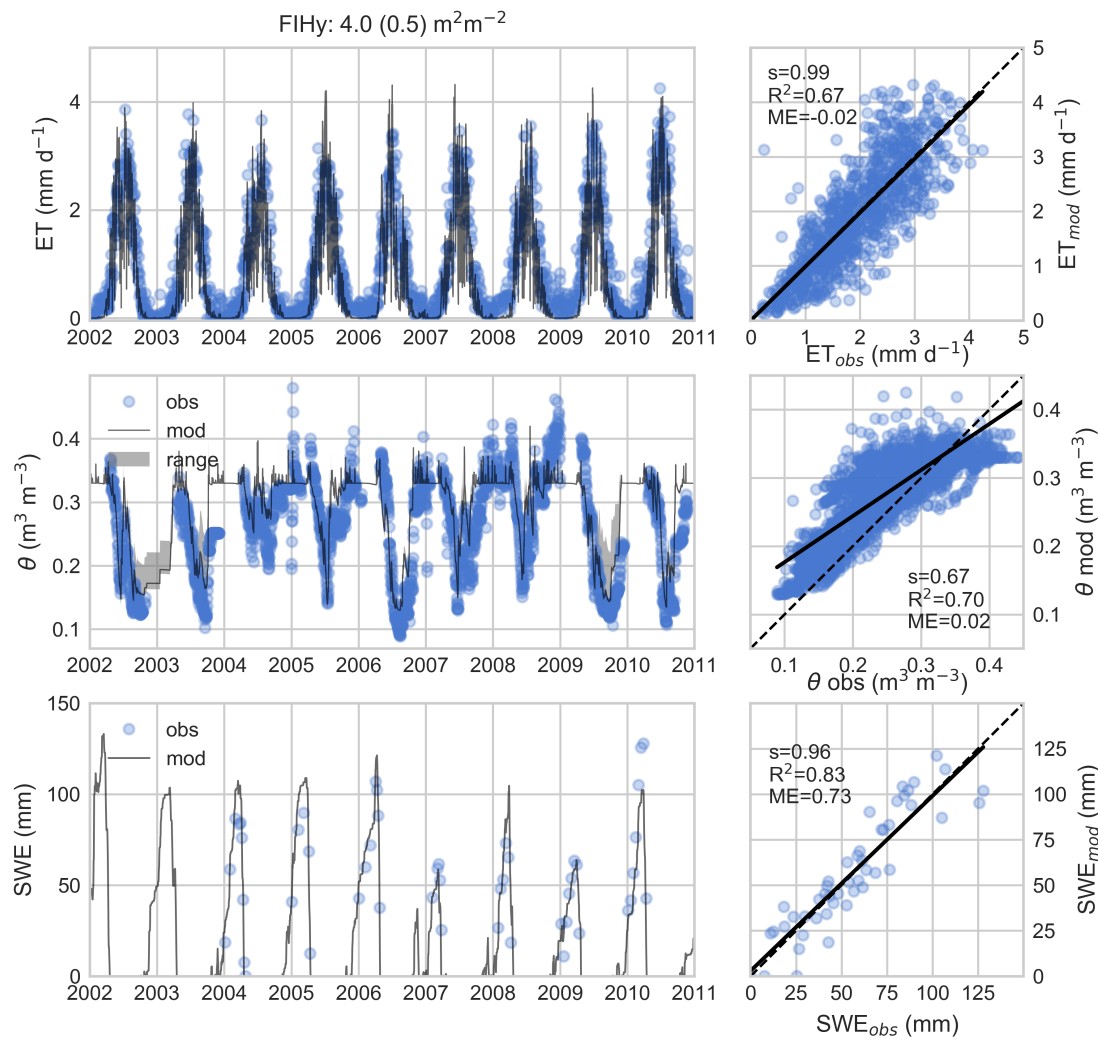

**Figure 3.** Modeled vs. measured dry-canopy ET at FIHy (top), root zone water content $\theta$ (middle) and snow water equivalent SWE (bottom). As soil freezing is not modelled, comparison of $\theta$ is restricted to conditions when soil temperature was $\geq 0°C$.





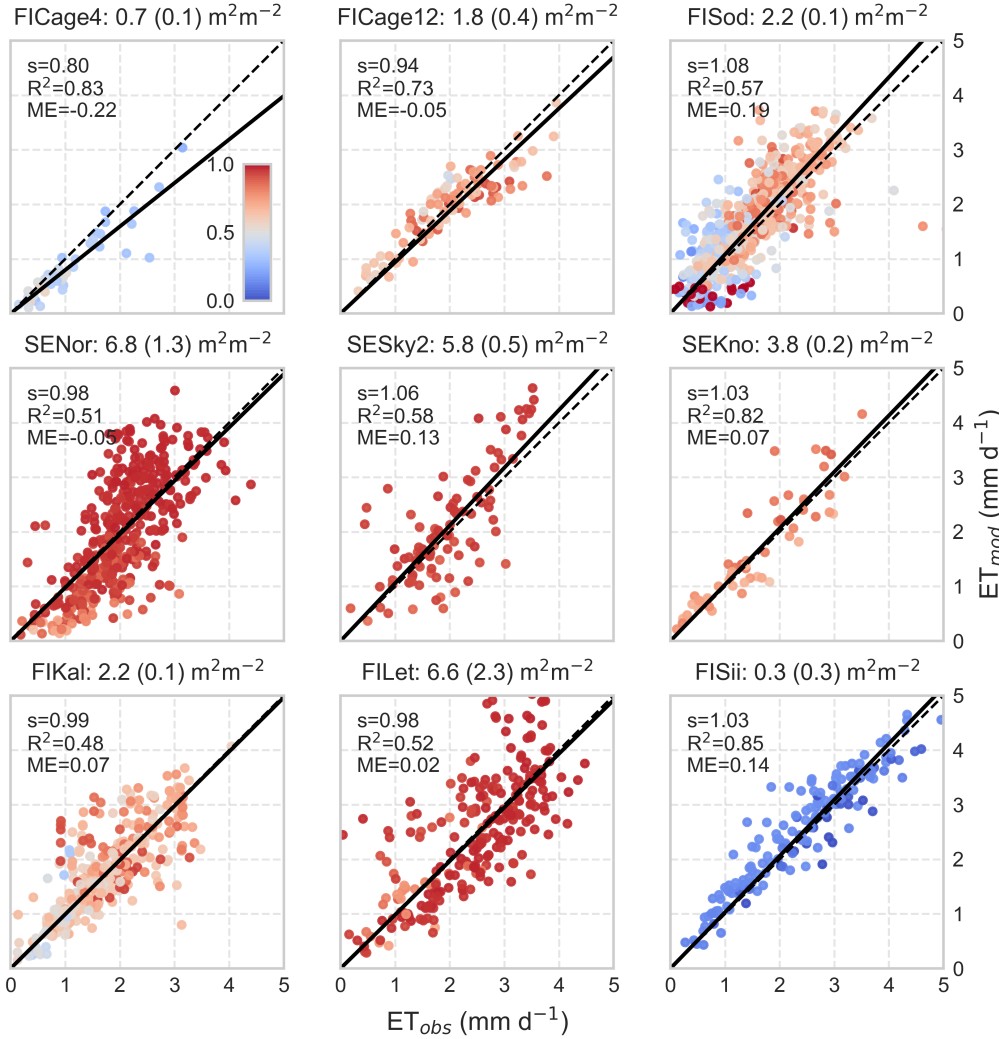

**Figure 4.** Scatter plots between modeled and observed daily stand-level ET during growing-season at the eddy-covariance flux sites in Finland and Sweden (Table 2). The title of each panel shows total LAI (maximum deciduous LAI in parenthesis). The slope $s$ and $R^2$ of linear regression forced through the origin and mean error ME are given and dashed line is the 1:1 line. Only dry canopy conditions, i.e. no rain during the day or previous day are included. At pristine fen peatland site FISii, $E_f$ was assumed non-limited by organic layer moisture. Color coding is according to transpiration to ET -ratio $T_r/(T_r + E_f)$.





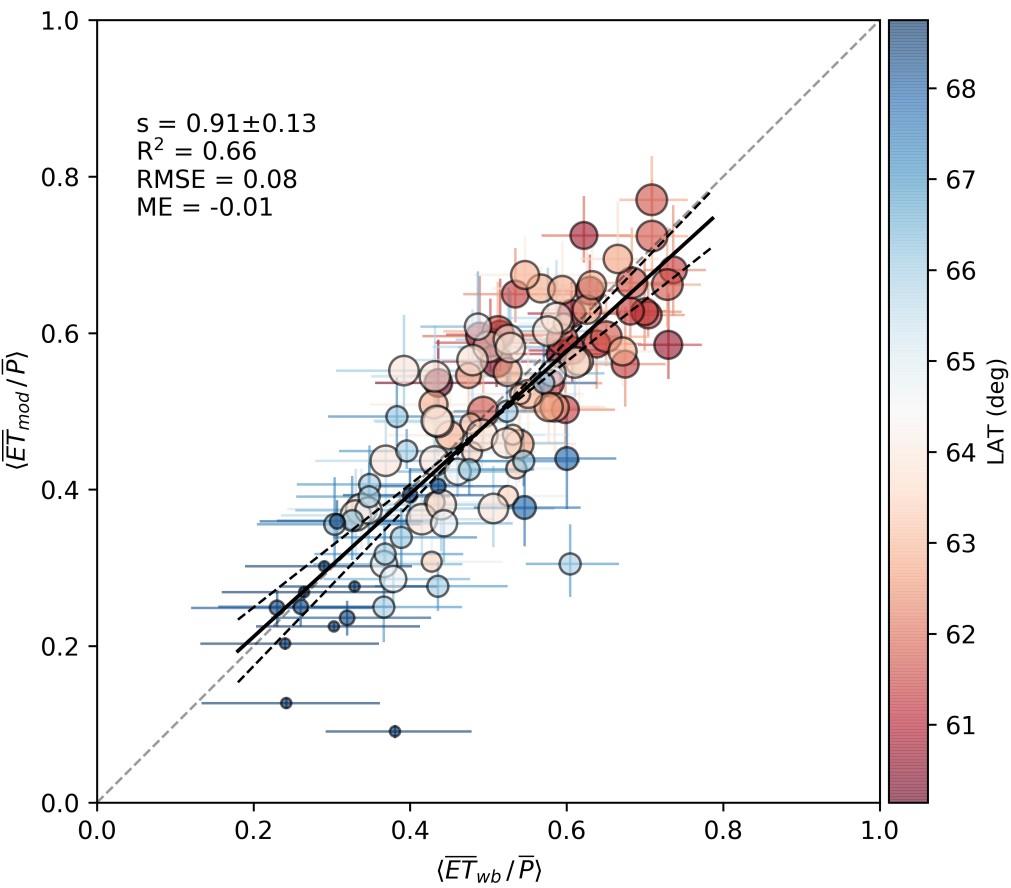

**Figure 5.** Modeled annual catchment evaporation fraction $\left\langle \overline{ET}_{mod}/\overline{P} \right\rangle$ compared to that inferred from catchment water balance $\left\langle \overline{ET}_{wb}/\overline{P} \right\rangle$. The vertical and horizontal errorbars show effect of parameter uncertainty and that of catchment area and $P$, respectively (see text). The colors refer to latitude and symbol size to catchment mean LAI (from 0.2 to 4.6 m$^2$m$^{-2}$). Using median year for each catchment (N=21), the respective statistics are: s 0.99±0.30, R$^2$ 0.67, RMSE 0.06, ME -0.003.





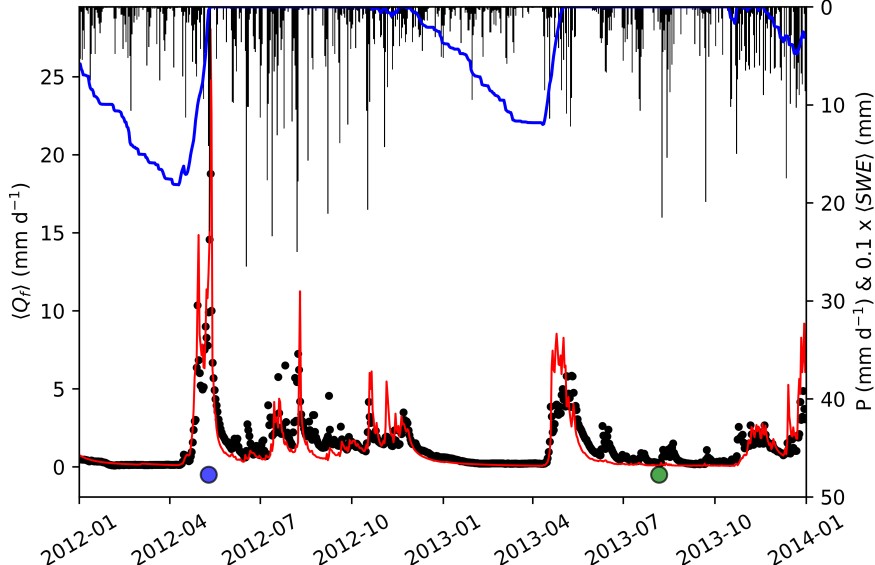

**Figure 6.** Measured (black) and modeled (red) specific discharge $Q_f$, daily precipitation $P$ (black bars) and mean snow water equivalent SWE (blue) for a wet (2012) and dry (2013) year at C3, Eastern Finland. Root zone moisture and *Topmodel* saturation deficit are show in Fig. 8 for the dates indicated by blue and green points.

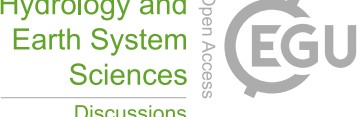



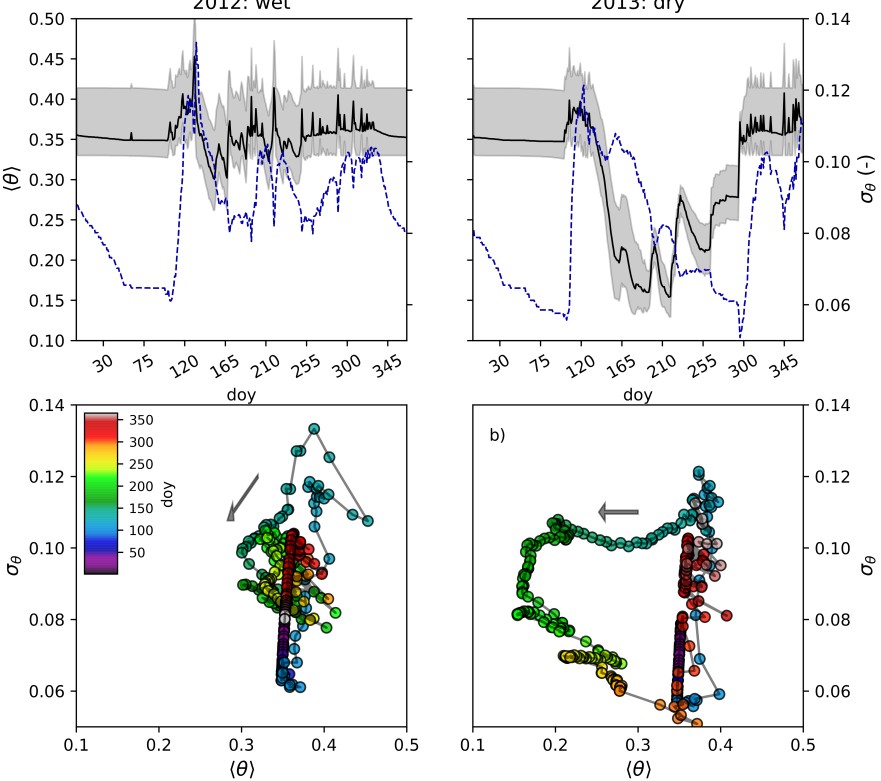

**Figure 7.** Temporal stability of soil moisture at C3 during wet year (2012) and dry year (2013). Top: Timeseries of catchment mean root zone moisture $\langle\theta\rangle$ ($m^3 m^{-3}$) and its spatial standard deviation $\sigma_\theta$ (dashed line). The gray range shows IQR of $\langle\theta\rangle$. Bottom: Relationship of $\sigma_\theta$ and $\langle\theta\rangle$ over the course of year.





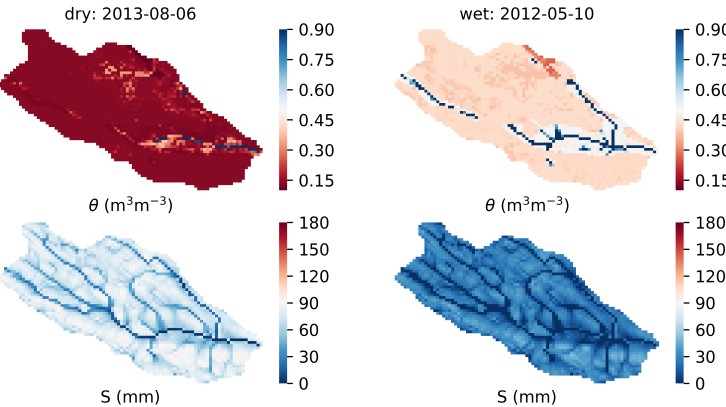

**Figure 8.** Snapshots of soil moisture patterns during wet and dry conditions. Water content $\theta$ in the root zone (top) and local saturation deficit $S$ of *Topmodel* (bottom).





**Figure 9.** Spatial variability of evaporation fraction $\overline{ET/P}$ and its components at C3 catchment in Eastern Finland from a long-term (2006-2016) run (left). The relationship of component fluxes interception evaporation $E$, transpiration $T_r$ and forest floor evaporation $E_f$ on LAI (right) is modified with spatial variability of soil type, proportion of deciduous trees and TWI.





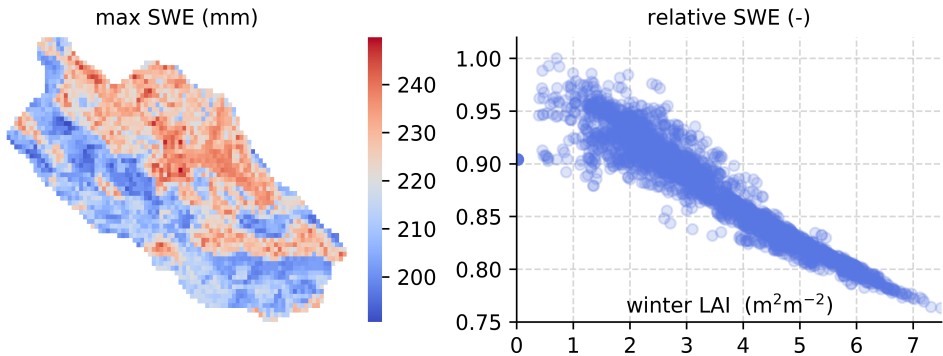

**Figure 10.** Spatial variability of maximum snow water equivalent (SWE) at C3 (left). The SWE relative to open area scales near-linearly with wintertime leaf-area index LAI (right).





**Table 1.** Generic parameter set used in stand and catchment scale simulations.

| parameter | value | units | explanation | Note |
|---|---|---|---|---|
| *Canopy* | | | | |
| $A_{max}$ | 10.0 | $\mu$ mol m$^{-2}$s$^{-1}$ | maximum leaf net assimilation rate | Suppl. material |
| $g_{1,c}$ | 2.1 | kPa$^{0.5}$ | stomatal parameter for conifers | shoot chamber in Launiainen et al. (2015) |
| $g_{1,d}$ | 3.5 | kPa$^{0.5}$ | stomatal parameter for deciduous | Lin et al. (2015), Suppl. material |
| $b$ | 50 | W m$^{-2}$ | half-saturation PAR of light response | Suppl. material |
| $k_p$ | 0.6 | - | radiation attenuation coefficient | Suppl. material |
| $r_w$ | 0.20 | - | critical relative extractable water | Lagergren and Lindroth (2002) |
| $r_{w,min}$ | 0.02 | - | minimum relative conductance | assigned |
| $G_f$ | 0.01 | m s$^{-1}$ | surface conductance for evaporation from wet forest floor | calibrated |
| $w_{max}$ | 1.5 | mm LAI$^{-1}$ | canopy storage capacity for rain | calibrated |
| $w_{max,snow}$ | 4.5 | mm LAI$^{-1}$ | canopy storage capacity for snow | Pomeroy et al. (1998); Essery et al. (2003) |
| $K_m$ | 2.5 | mm d$^{-1}$ | melt coefficient in open area | Kuusisto (1984) |
| $K_f$ | 0.5 | mm d$^{-1}$ | freezing coefficient | |
| *Bucket* | | | | |
| $z_{s,org}$ | 0.05 | m | organic layer depth | |
| $\theta_{s,org}$ | 0.90 | - | porosity of org. layer | |
| $\theta_{fc,org}$ | 0.30 | - | field capacity of org. layer | |
| $\theta_{crit,org}$ | 0.24 | - | critical vol. water content of org. layer | |
| $z_s$ | 0.4 | m | root zone depth | |
| $\theta_s$ | | m$^3$m$^{-3}$ | porosity of root zone layer | |
| $\theta_{fc}$ | | m$^3$m$^{-3}$ | field capacity of root zone layer | |
| $\theta_{wp}$ | | m$^3$m$^{-3}$ | wilting point of root zone layer | |
| $K_{sat}$ | | m s$^{-1}$ | saturated hydraulic conductivity | |
| $\beta$ | | - | decay parameter of hydraulic conductivity | |
| *Topmodel* | | | | |
| $T_o$ | 0.001 | m s$^{-1}$ | transmissivity at saturation | assigned |
| $m$ | Catchment specific | m | effective soil depth | calibrated against discharge |





**Table 2.** Eddy-covariance sites used in stand-scale validation

| Site | Name | Lat Lon | Site type | LAI ($m^2 m^{-2}$) | Height (m) | Age (yr) | P (mm) | Ta (C) | Years | Soil | Reference |
|---|---|---|---|---|---|---|---|---|---|---|---|
| FIHy | Hyytiälä | 61.85N, 24.30E | Scots pine, mineral soil | 4.0 (0.5) | 15 (2) | 45 (5) | 709 | 2.9 | 2000-2005 | medium texture* | Hari and Kulmala (2005), Launiainen et al. (2015) |
| FICage4 | Hyytiälä 4yr | 61.85N, 24.30E | Scots pine, mineral soil | 0.6 (0.1) | 0.4 (0.2) | 4 (1) | 709 | 2.9 | 2007-2010 2000 | medium texture | Rannik et al. (2002), Kolari et al. (2004) |
| FICage12 | Hyytiälä 12yr | 61.85N, 24.30E | Scots pine, mineral soil | 1.8 (0.4) | 1.7 | 12 (1) | 709 | 2.9 | 2002 | medium texture | Kolari et al. (2004) |
| FISod | Sodankylä | 67.36N, 26.64E | Scots pine, mineral soil | 2.2 (0.2) | 13 (1) | 150 (50) | 527 | -0.4 | 2001-2009 | coarse texture | Aurela (2005), Thum et al. (2007) |
| FIKal | Kalevansuo | 60.65N, 23.96E | Drained peatland | 2.5 (0.5) | 15 (1) | 40 (3) | 606 | 4.6 | 2005-2008 | peat | Lohila et al. (2011) |
| FILet | Lettosuo | 60.64N, 23.96E | Drained peatland | 6.6 (2.3) | 20 (2) | 40 (3) | 627 | 4.6 | 2010-2012 | peat | Koskinen et al. (2014) |
| SENor | Norunda | 60.09N, 17.48E | Mixed coniferous, mineal soil | 6.5 (1.3) | 27 (3) | 100 (20) | 527 | 5.5 | 1996, 1997, 2003 | medium texture | Lundin et al. (1999), Lindroth et al. (2008) |
| SEKno | Knottåsen | 61.0N, 16.22E | Norway spruce, mineral soil | 3.6 (0.2) | 16.5 (1) | 40 (3) | 613 | 3.4 | 2006, 2009 | medium texture | Berggren et al. (2004), Lindroth et al. (2008) |
| SESky2 | Skyttorp 2 | 60.13N, 17.84E | Scots pine, mineral soil | 5.3 (0.5) | 15.8 (1) | 39 (2) | 527 | 5.5 | 2005 | medium texture | Gioli et al. (2004) |
| FISii | Siikaneva | 61.85N, 24.30E | Boreal fen | 0.3 (0.3) | 0.3 | | 709 | 2.9 | 2011, 2013, 2015 | peat | Alekseychik et al. (2017) |

LAI is ecosystem 1-sided leaf-area index (deciduous LAI in parenthesis); $P$ is annual mean precipitation; $T_a$ mean annual air temperature and soil type refers to Table 2 in the main document.

* for runs at FIHy, site-specific field capacity $\theta_{fc} = 0.3$ and wilting point $\theta_{wp} = 0.1$ corresponding to main root zone (Launiainen et al., 2015) were used.



**Table 3.** Parameters and their ranges used in the global sensitivity analysis (Morris method) at stand scale.

| canopy parameters | range | unit | explanation |
|---|---|---|---|
| $LAI$ | 0.1 - 8.0 | $m^3\,m^{-3}$ | total leaf area index |
| $f_d$ | 0.0 - 1.0 | - | deciduous fraction |
| $g_1$ | 1.0 - 7.0 | - | stomatal parameter |
| $A_{max}$ | 6.0 - 14.0 | $\mu mol\,m^{-2}$ (leaf) $s^{-1}$ | maximum leaf net assimilation rate |
| $b$ | 20.0 - 60.0 | $W\,m^{-2}$ | half-saturation PAR of light response |
| $k_p$ | 0.4 - 0.6 | - | radiation attenuation coefficient |
| $r_w$ | 0.05 - 0.50 | - | critical relative extractable water |
| $G_f$ | $1.0\times10^{-3}$ - $1.0\times10^{-1}$ | $m\,s^{-1}$ | surface conductance for evaporation from wet forest floor |
| $h_c$ | 1.0 - 30.0 | m | canopy height |
| $w_{max}$ | 0.5 - 3.0 | $mm\,LAI^{-1}$ | canopy storage capacity for rain |
| $w_{max,snow}$ | 1.0 - 10.0 | $mm\,LAI^{-1}$ | canopy storage capacity for snow |
| bucket parameters | | | |
| $z_s$ | 0.2 - 0.7 | m | root zone depth |
| $z_{s,org}$ | 0.02 - 0.1 | m | organic layer depth |
| $\theta_{fc,org}$ | 0.2 - 0.4 | $m^3\,m^{-3}$ | field capacity of org. layer |
| $\theta_{crit,org}$ | 0.1 - 0.4 | $m^3\,m^{-3}$ | critical vol. water content of org. layer |



**Table 4.** Sensitivity of *Canopy* sub-model predictions to parameter variability. Mean ($\mu$) and standard deviation ($\sigma$) of the distribution of elementary effects for evapotranspiration (ET), transpiration ($T_r$), evaporation from canopy interception (E), ground evaporation ($E_f$), and snow water equivalent (SWE). Negative sign of $\mu$ indicate output variable decreases when parameter value increases. Units are in mm a$^{-1}$.

| Parameters | ET $\mu$ | ET $\sigma$ | $T_r$ $\mu$ | $T_r$ $\sigma$ | E $\mu$ | E $\sigma$ | $E_f$ $\mu$ | $E_f$ $\sigma$ | SWE $\mu$ | SWE $\sigma$ |
|---|---|---|---|---|---|---|---|---|---|---|
| $LAI$ | 100 | 230 | 240 | 280 | 140 | 94 | -130 | 77 | -36 | 33 |
| $f_d$ | -9.5 | 13 | -20 | 16 | -32 | 14 | 11 | 8.3 | 23 | 21 |
| $g_1$ | 97 | 82 | 97 | 82 | 0.0 | 0.0 | -0.0 | +0.0 | 0.0 | 0.0 |
| $A_{max}$ | 56 | 38 | 56 | 38 | 0.0 | 0.0 | -0.0 | 0.1 | 0.0 | 0.0 |
| $b$ | -42 | 23 | -42 | 23 | 0.0 | 0.0 | +0.0 | +0.0 | 0.0 | 0.0 |
| $k_p$ | -3.3 | 13 | 20 | 19 | 0.2 | 0.1 | -23 | 15 | -0.2 | 0.1 |
| $r_w$ | -7.3 | 6.6 | -7 | 6.6 | 0.0 | 0.0 | 0.0 | 0.0 | 0.0 | 0.0 |
| $G_f$ | 17 | 52 | -2.2 | 5.9 | 0.0 | 0.0 | 19 | 53 | 0.0 | 0.0 |
| $h_c$ | -8.2 | 29 | -9.0 | 2.0 | 20 | 24 | 0.7 | 13 | -4.4 | 4.9 |
| $w_{max}$ | -22 | 22 | -19 | 23 | 69 | 43 | -2.7 | 2.3 | 5.7 | 4.4 |
| $w_{max,snow}$ | -0.9 | 2.0 | -1.9 | 2.3 | 11 | 14 | 1.0 | 0.8 | -28 | 21 |
| $z_s$ | 58 | 44 | 58 | 44 | 0.0 | 0.0 | -0.1 | 0.3 | 0.0 | 0.0 |
| $z_{s,org}$ | 19 | 24 | -3.2 | 5.6 | 0.0 | 0.0 | 22 | 24 | 0.0 | 0.0 |
| $\theta_{fc,org}$ | 7.8 | 8.5 | -1.1 | 2.0 | 0.0 | 0.0 | 8.9 | 8.5 | 0.0 | 0.0 |