# Peer review of "Modeling boreal forest evapotranspiration and water balance at stand and catchment scales: a spatial approach"

_Hydrology and Earth System Sciences, 2019_

## Referee Comment (RC1) · Anonymous Referee #1 · 7 Mar 2019

**General comments**

The paper introduces a model for "upscaling evaporation and other hydrological processes from grid cell to a catchment level". The manuscript is well written, well-structured, and comprehensive, including parameter sensitivity analyses and evaluation against different sites and hydrological flows.

The motivation for the study is stated in the introduction as: (1) "to improve evaporation description by more physiologically-based approach", and (2) to take full advantage of open spatial data. In relation to this, the only substantial revisions I would like to propose is to (1) compare the simulation results from the "physiologically-based approach"

with an "empirical and simple approach" and (2) compare the simulation results from a model run that makes use of the detailed GIS data and one that does not. I think such analyses would help the reader better relate the results to the motivations stated, and better understand the added-value of the presented "improved approach".

**Specific comments**

Symbols: Please consult the HESS manuscript guidelines Symbols and equations. E.g., (b) "Multi-letter variables should be avoided. Instead use single-letter variables with subscript (e.g. $E_{\mathrm{RMS}}$ instead of RMSE... "

Terminology: Please consider using evaporation for total evaporation, instead of evapotranspiration (Savenije, 2004).

Names of sites: Please consider using a naming system that is easier to remember and relate to for the reader. E.g., FIHy-Pine instead of just FIHy.

Figures: Please consider adding letters (and sometimes subtitles) to subplots for easier identification. Please add legend to the figures and not only explain in caption (e.g., Fig 7).

Title: Please consider adding "boreal" to the title to be more precise, i.e., "Modeling boreal forest...."

P2L8-L19:

Please consider providing a more balanced review of the different types of models that recognize more merits and disadvantages of the different model types. The reference (Reed et al., 2004) cited, for example found lumped models to have a "better overall performance than distributed models" and (Winsemius et al., 2006). It would also be worth adding a coupled of sentenced on the models that integrate combine lumped features with distributed physically based ones (Gao et al., 2013; Khakbaz et al., 2012), particularly since the study itself presents a semi-distributed solution. The statement at P2L13 "problematic for ungauged catchments" also leaves the impression

that lumped approaches are not relevant for ungauged basins, while for example (Hrachowitz et al., 2013; Hundecha et al., 2008; Winsemius et al., 2009) do not dismiss their relevance in such contexts. (Hrachowitz et al., 2013) for example states: "During the PUB Decade, an increasing understanding of the importance of openness towards different approaches, and the willingness to communicate and search for opportunities developed."

Please also note that distributed models can have conceptual components, and are not always necessarily "physically-based" (as implied by the sentence formulation at P2L10).

Please also consider placing the models used in the study in context with other model types reviewed in the introduction.

P2L33 "Penman-Monteith equation": Please consider citing the original reference (Monteith, 1965).

P3L9 "to derive a model": I am not sure I understand the word choice "derive" here, do you simply mean "used in"?

P5L16: Please explain the threshold parameter and refer to where its values can be found.

P5L18 Eq 7: Please consider adding reference to the equation. Is it (van Genuchten, 1980)?

P5L20: Please write out the equation (?). In general, please consider referring to Table 3, each first time the parameters listed are mentioned.

P10L24: Please consider putting Fig S1 in the main manuscript.

P1213: Please consider referring back to the methods section.

P16L12-13: Please note that methods for determining root zone storage capacity using satellite based information has been developed e.g., (Gao et al., 2014; Wang-

[Figure]

Erlandsson et al., 2016).

P.19L1- : Please consider discussing a bit more in detail in what way the model is advantageous for the applications listed.

P31 Caption: How many days are included/excluded? How much of the mean annual evaporation is taken into account when considering only the dry canopy conditions?

P34 Fig 7: Please spell out IQR. Should the right axis of subplot b be blue? Please add units.

P33Fig 6: Please consider including a measure of the runoff performance? And present the measure for all catchments?

Fig 35: Pleas state the catchment name considered.

P38: Please provide justification, explanation or reference for of all values selected.

**References**

Gao, H., Hrachowitz, M., Fenicia, F., Gharari, S., Savenije, H.H.G., 2013. Testing the realism of a topography driven model (FLEX-Topo) in the nested catchments of the Upper Heihe, China. Hydrol. Earth Syst. Sci. Discuss. 10, 12663–12716. https://doi.org/10.5194/hessd-10-12663-2013

Gao, H., Hrachowitz, M., Schymanski, S.J., Fenicia, F., Sriwongsitanon, N., Savenije, H.H.G., 2014. Climate controls how ecosystems size the root zone storage capacity at catchment scale. Geophys. Res. Lett. 41, 7916–7923. https://doi.org/10.1002/2014GL061668

Hrachowitz, M., Savenije, H.H.G., Blöschl, G., Mcdonnell, J.J., Sivapalan, M., Pomeroy, J.W., Arheimer, B., Blume, T., Clark, M.P., Ehret, U., Fenicia, F., Freer, J.E., Gelfan, A., Gupta, H. V, Hughes, D.A., Hut, R.W., Montanari, A., Pande, S., Tetzlaff, D., Troch, P.A., Uhlenbrook, S., Wagener, T., Winsemius, H.C., Woods, R.A., Zehe, E., Cudennec, C., Savenije, H.G., 2013. A decade of Predictions in Ungauged Basins (PUB)-a review.

Hydrol. Sci. J. 58, 1198–1255. https://doi.org/10.1080/02626667.2013.803183 Hundecha, Y., Ouarda, T.B.M.J., Bárdossy, A., 2008. Regional estimation of parameters of a rainfall-runoff model at ungauged watersheds using the "spatial" structures of the parameters within a canonical physiographic-climatic space. Water Resour. Res. 44. https://doi.org/10.1029/2006WR005439

Khakbaz, B., Imam, B., Hsu, K., Sorooshian, S., 2012. From lumped to distributed via semi-distributed: Calibration strategies for semi-distributed hydrologic models. J. Hydrol. 418–419, 61–77. https://doi.org/10.1016/J.JHYDROL.2009.02.021

Monteith, J.L., 1965. Evaporation and environment, in: Symp Soc Exp Biol. Cambridge University Press, Swansea, pp. 205–234.

Reed, S., Koren, V., Smith, M., Zhang, Z., Moreda, F., Seo, D.-J., DMIP Participants, and, 2004. Overall distributed model intercomparison project results. J. Hydrol. 298, 27–60. https://doi.org/10.1016/J.JHYDROL.2004.03.031

Savenije, H.H.G., 2004. The importance of interception and why we should delete the term evapotranspiration from our vocabulary. Hydrol. Process. 18, 1507–1511. https://doi.org/10.1002/hyp.5563

Schymanski, S.J., Or, D., 2017. Leaf-scale experiments reveal an important omission in the Penman–Monteith equation. Hydrol. Earth Syst. Sci. 21, 685–706. https://doi.org/10.5194/hess-21-685-2017

van Genuchten, M.T., 1980. A Closed-form Equation for Predicting the Hydraulic Conductivity of Unsaturated Soils. Soil Sci. Soc. Am. J. 44, 892–898. https://doi.org/10.2136/sssaj1980.03615995004400050002x

Wang-Erlandsson, L., Bastiaanssen, W.G.M., Gao, H., Jägermeyr, J., Senay, G.B., Van Dijk, A.I.J.M., Guerschman, J.P., Keys, P.W., Gordon, L.J., Savenije, H.H.G., 2016. Global root zone storage capacity from satellite-based evaporation. Hydrol. Earth Syst. Sci. 20. https://doi.org/10.5194/hess-20-1459-2016

Winsemius, H.C., Savenije, H.H.G., Gerrits, A.M.J., Zapreeva, E.A., Klees, R., 2006. Comparison of two model approaches in the Zambezi river basin with regard to model reliability and identifiability. Hydrol. Earth Syst. Sci. 10, 339–352. https://doi.org/10.5194/hess-10-339-2006

Winsemius, H.C., Schaefli, B., Montanari, A., Savenije, H.H.G., 2009. On the calibration of hydrological models in ungauged basins: A framework for integrating hard and soft hydrological information. Water Resour. Res. 45, W12422. https://doi.org/10.1029/2009WR007706

―――――――――――――――――――――――

---

## Referee Comment (RC2) · Anonymous Referee #2 · 14 Mar 2019

Comments for hss-2019-45: Modeling forest evapotranspiration and water balance at stand and catchment scales: a spatial approach by Launiainen et al.

The authors presented a hydrological model combining processes from canopy to top soil and below, and validation of the model performance for spatially-averaged ET (and its components), moisture, SWE and discharge was carried out for both forest stands and catchments. The usefulness of enriched open GIS database of soil, vegetation and land use etc. was particularly mentioned. The advantages of the model are that it incorporates the available dataset to well reproduce the major hydrological measurements almost at all forests and catchments at a fairly high resolution, and it proves that

the parameter transferability from stand to catchment scales is possible. Although the high sensitivity of ET and its components to LAI is not surprising. In conclusion, the work is worth doing and publishing.

With these said, the m/s really gave me headache to read it through and read it again, because it is badly written in terms of flow of logic, structure of the subsections, and confusion of main points for discussion. It seems the authors have a chaotic design for the m/s, for example when Fig 3 is talked about in page 12 the authors suddenly refers to Fig 10, and then jump back to Fig 4; another example is on page 14 about soil moisture variability description, the fig 7- fig 8- fig7 loop. I understand these processes have connections but it is the authors' responsibility to make the connections in a clear order. I think the manuscript needs substantial rework before it can be published.

Specific comments for the authors' reference: Page 4 Line 25 and 28: in the two equations, two symbols for the same leaf area index are used.

Page 5 in the eq.6, describe how the threshold parameter is determined, based on measurements or calibrated? Line 20: since fs is an important stress function in eq.4, it would be better to explicitly write it out, otherwise the readers have to find it by themselves in the literature cited. Note also the math express: at some places you used the form of y= exp(x), while at others you used y=ex. Be consistent.

Page 7: symbol beta is used in both eq.16 and eq.19, but with different meanings.

Page 9 Line 8-10: delete – it is a repetition of page 8 Line 18-20. Also write in the 2.6.2 section clearly what a larger $\mu$ and $\sigma$ infers, otherwise when it comes to Results 3.1 and the relevant table, it is hard to comprehend.

Page 11: The section 2.6 is 'Model validation at stand and catchment scales', but 2.6.3 is GIS preprocessing, and 2.6.4 is calibrating Topmodel. I don't think they are appropriately positioned. 2.6.3 is better fitted in Model input section, and 2.6.4 merged with parameter sensitivity section.
Page 14: paragraph starting from Line 4 can be written better by reorganizing a few sentences. The last sentence is important but the plot is not shown. Better show at least an example in one sub-slot either within the plot or next to the plot in Fig 6. Since this will be an important support for your argument.

Page 14: 3.4.1- This part does not read well. I see you want to explain the temporal and spatial variability of soil moisture, and relate the variability with drainage and/or ET and its components. However, the current writing mixes them badly. Suggestion: describe Fig 7 well first for temporal variability, and Fig 8 second for spatial variability. For Fig 9 because you talk about it in the next section, so I suggest not to mention it here. Just describe the plots. All explanations can be moved to Discussion. Line 27-28: Fig 9 is the long-term averaged values, cannot tell anything in between these years, like doy 180. Line 30 – to support the rainfall effect, you must plot the rainfall bars in Fig 7.

Page 15: paragraph about SWE - move SWE to the last paragraph, i.e. describe fig 9 before fig 10, avoid mix them for mind-jumping

Page 16: Discussion – it is difficult to follow and digest, simply because it was badly organized. Currently it gives the impression of no logic of flow. Confusion is caused about what the main points are under discussion. Apparently, the proposed model has the capability to simulate hydrological processes across stand and catchment scales, and sensitivity analysis shows it only has a limited number of parameters significantly influencing the modeling results; Open GIS database application in hydrological model is also mentioned and discussed. Some features that the model has not developed are mentioned in Discussion which I think is unnecessary; and the reason why spatial validation of moisture, ET or SWE should be mentioned in Method validation section, not here. In one word, I would like the authors to think carefully about what they want to discuss or what are the main take-home messages they want to readers to get?

Page 18: first paragraph in 4.1.1 can be deleted. Part of the potential applicability of

the model has already been mentioned in the previous paragraphs.

I hope the authors notice that there are many NOT SHOWN in the m/s, which makes the m/s sound incomplete.
* * *

---

## Author Response (AR1)

**Response to Editor Comments**

We respond below to editor and reviewer comments. The changes in the manuscript are detailed using track-changes at the end of this document.

Comments to the Author:

After reading the manuscript and the 2 review reports, I noticed that there are some conflicting suggestions, mainly about the structure of the manuscript. Reviewer #1 agrees with the structure, while Reviewer #2 advices to improve it. After reading the manuscript, I think the general structure of the manuscript is OK, although here and there some minor improvements can be made (see specific suggestions both reviewers). However, I also noticed that the scope of the manuscript is not so clear, which might led to the problems that reviewer #2 experienced. Also to me, it is not entirely clear what the scope of the paper is. Is it the development of a new model (then also better explain the differences with existing models and why those ones are not good enough)? Is it investigating the use of open data (what is so special about using open data? Closed or open source, the data in it self is not different right? and if so, please explain why it's special)? Or is it finding the main drivers of ET-partitioning? Or something else? Please be more specific in the final paper. This main objective can also be emphasized in the title of the manuscript.

Furthermore, I do agree with the authors to ignore the model comparison as suggested by Reviewer #1. However, note that this request likely came from an unclear study scope.

Response: We agree with the Editor opinion that study scope was a bit unclear and have now tried to clarify it.

Lastly, I do agree with the suggestion of reviewer#1 to change the catchments names to more descriptive ones. And also carefully check the units, because some errors exists e.g.,:

- P4L5: ET is here in m/s (and not mm/d, otherwise multiply equation with 1000 and 86400s)

- Eq 2-4: mixing micro-mol (A) with mol (Cair) is rather dangerous in empirical equations.

Response: Units were checked throughout and corrected whenever necessary. The catchment name is now mentioned in the text whenever referring to a specific catchment. Regards to eddy-covariance site names; we keep the original ones but refer in the text to dominant species, e.g. FIHy (pine), when it improves the clarity.

We revised variable names etc. according to HESS guidelines; however for clarity we keep multi-letter variables ET (evapotranspiration), SWE (snow water equivalent), LAI (leaf-area index) and TWI (topographic wetness index) as they are well-defined abbreviations.

Minor others:

-P6L7: what is the use of "mm=kg/m2"? Please use conversion through density of water/surface area

R: done

-P8L13: Qr=-S: How is this possible? Q has dimension L/T and S dimension L. So they can not be the same. Please add dt in equation.

R: thanks for pointing out the error; it is now corrected.

-P9L14: How is it possible that rainfall interception is dependent on the dt?

R: this is related to (calibrated) interception capacity parameter that is dependent on model timestep. As we consider in this work daily timestep and calibrate Wmax, we removed this unclear sentence.

-P9L19: is 100 iterations for a monte carlo simulation not too small?

R: Here only one or two parameters within a narrow initial range were varied so this was sufficient

-Fig 2: add xy-labels plus units

-Fig 3: change y-label into ET_mod, to be consistent with right hand side figure

R: both corrected accordingly

-Table4: the units SWE is not mm/a as written in the caption. Please correct.

R: corrected

Anonymous Referee #1

**General comments**

The paper introduces a model for "upscaling evaporation and other hydrological processes from grid cell to a catchment level". The manuscript is well written, well structured, and comprehensive, including parameter sensitivity analyses and evaluation against different sites and hydrological flows.

The motivation for the study is stated in the introduction as: (1) "to improve evaporation description by more physiologically-based approach", and (2) to take full advantage of open spatial data. In relation to this, the only substantial revisions I would like to propose is to (1) compare the simulation results from the "physiologically-based approach" with an "empirical and simple approach" and (2) compare the simulation results from a model run that makes use of the detailed GIS data and one that does not.

I think such analyses would help the reader better relate the results to the motivations stated, and better understand the added-value of the presented "improved approach".

**Response**

We are grateful for the thorough review, excellent comments and concrete suggestions how to further improve the manuscript.

Regarding to general comment (1) on comparing the proposed 3-source ET-model to empirical and simple approach our response is as follows:

The level of detail evapotranspiration is necessary to describe must depend on the goal and application. For predicting ET at a single site, or to get reasonable partitioning of long-term water balance at catchment scale, in principle any method embedding the main drivers (radiation, temperature and air humidity) could be used with proper calibration / parameterization. The main aim of the study is, however, seek for alternative approach where physical and physiological arguments are coupled as a relatively simple scheme then shown to be applicable at range of forest/peatland sites across latitudinal climate gradient in Finland/Sweden. Thus, we expect the proposed model can reasonably predict the spatial heterogeneity of catchment / landscape ET, as it can utilize open spatial data available. This is crucial for the second motivation of the study. To summarize, we see the point of the reviewer but in our opinion extending the paper towards model comparison would distract the reader from the main message of the study. To make the above argument more clear, we check the Introduction and Discussion for this regard.

The general comment (2) on studying the importance of spatial heterogeneity on simulation results is a good point, and one of the main motivations of developing the proposed model framework. We, however, opted against adding specific analysis of the role of different sources of spatial heterogeneity to keep the manuscript length tolerable. It was not specifically mentioned that which model outputs would be required to analyze more deeply but regarding to water budget components and soil moisture variability these are already partly included in:

- Fig. 9 showing the role of spatial heterogeneity of vegetation, soil type and topographic wetness index on long-term evapotranspiration and its partitioning;
- Fig 8. Showing how root zone moisture varies spatially at two extreme cased (moist / dry)
- Fig 7: Temporal evolution of mean soil moisture and its variance, and discussion around.

Interpreting these figures with the spatial data layers (Fig. 2 in original MS) will allow interested reader to gain some insights on the role of spatial heterogeneities on respective model outputs.

The importance of spatial landscape heterogeneity on catchment water balance partitioning at annual, and on specific discharge at daily timescale could be further studied by setting up a factorial model experiment e.g. at catchment C3 used for illustrations. One can consider four spatial layers (LAI, deciduous fraction, soil type, TWI) as independent variables, each with two levels (0 = constant, 1 = spatially variable) yielding to $2^4$ = 16 combinations. This analysis could be added into Supplementary material but we would prefer a separate study.

**Specific comments (C) and our responses (R)**

C: Symbols: Please consult the HESS manuscript guidelines Symbols and equations. E.g., (b) "Multi-letter variables should be avoided. Instead use single-letter variables with subscript (e.g. ERMS instead of RMSE. . . "

R: Revised to fulfill HESS manuscript guidelines

C: Terminology: Please consider using evaporation for total evaporation, instead of evapotranspiration (Savenije, 2004).

R: We are aware of Savenije (2004) - paper and share its opinion on importance of distinguishing between different components of evaporative water fluxes as they have partially different controls. This is where the proposed scheme is beneficial compared to most of the previous models. Being aware that terminology often differs between disciplines, evapotranspiration is well-recognized and well-defined term at least in ecohydrology, atmospheric sciences etc. so we prefer to keep the original terminology.

C: Names of sites: Please consider using a naming system that is easier to remember and relate to for the reader. E.g., FIHy-Pine instead of just FIHy.

R: We acknowledge the comment but as the main point of the manuscript is to show general applicability of the model across sites, specific sites are of less importance. Therefore, no changes are made to site names but the dominant species is now mentioned in text whenever we think it improves the clarity.

Figures: Please consider adding letters (and sometimes subtitles) to subplots for easier identification. Please add legend to the figures and not only explain in caption (e.g., Fig 7).

R: We aimed to ease interpretation when preparing the final figures. Please note that Fig 6 and 7 are changed substantially to improve the clarity.

Title: Please consider adding "boreal" to the title to be more precise, i.e., "Modeling boreal forest. . .."

R: We considered the comment and well understand the point. However, as the model approach is not limited to boreal forests (except for some of the specific parameters) we prefer the original title.

C: Please consider providing a more balanced review of the different types of models that recognize more merits and disadvantages of the different model types. The reference (Reed et al., 2004) cited, for example found lumped models to have a "better overall performance than distributed models" and (Winsemius et al., 2006). It would also be worth adding a coupled of sentenced on the models that integrate combine lumped features with distributed physically based ones (Gao et al., 2013; Khakbaz et al., 2012), particularly since the study itself presents that lumped approaches are not relevant for ungauged basins, while for example (Hrachowitz et al., 2013; Hundecha et al., 2008; Winsemius et al., 2009) do not dismiss their relevance in such contexts. (Hrachowitz et al., 2013) for example states: "During the PUB Decade, an increasing understanding of the importance of openness towards different approaches, and the willingness to communicate and search for opportunities developed."

Please also note that distributed models can have conceptual components, and are not always necessarily "physically-based" (as implied by the sentence formulation at P2L10).

Please also consider placing the models used in the study in context with other model types reviewed in the introduction.

R: Thanks for pointing out these biases and unintended messages given in the section. The excellent suggestions of the reviewer will be embedded into the revised Introduction that now is more balanced.

C:P2L33 "Penman-Monteith equation": Please consider citing the original reference (Monteith, 1965).

R: References provided discuss some of the current methods used for ET in hydrological models, and were not meant refer to Penman-Monteith equation. Sentence was reformulated.

C: P3L9 "to derive a model": I am not sure I understand the word choice "derive" here, do you simply mean "used in"?

R: "to derive" changed to "to develop"

C: P5L16: Please explain the threshold parameter and refer to where its values can be found.

R: It was thought role of threshold parameter in linear eq. 6 is self-explanatory; however as this was also pointed out by Reviewer 2 we explain its meaning in text and refer to Table 1. The value used here is based on Lagergren and Lindroth (2002) sap-flow study on Scots pine and Norway spruce, as cited in P5L14.

C: P5L18 Eq 7: Please consider adding reference to the equation. Is it (van Genuchten, 1980)?

R: The concept of plant available water or relatively extractable water (eq. 7) has been widely used in the literature; we will provide original reference if possible.

C: P5L20: Please write out the equation (?). In general, please consider referring to Table 3, each first time the parameters listed are mentioned.

R: We provide equation and refer to Table 1 for parameter values and their sources. Please note that Table 3 provides parameter ranges for sensitivity analysis.

C: P10L24: Please consider putting Fig S1 in the main manuscript.

R: Improved version of Fig S1 is be added to the main manuscript as suggested.

C: P1213: Please consider referring back to the methods section.

R: The remark will be considered when finalizing the revised manuscript.

C: P16L12-13: Please note that methods for determining root zone storage capacity using satellite based information has been developed e.g., (Gao et al., 2014; Wang-Erlandsson et al., 2016).

R: Thanks for pointing out these excellent references; remark to possibility of using remote-sensing methods to determine root zone storage capacity was added.

C: P.19L1- : Please consider discussing a bit more in detail in what way the model is advantageous for the applications listed.

R: The comment was considered but to avoid repetition from earlier discussion we decide not to expand this section. Reviewer 2 asked to remove this section.

C: P31 Caption: How many days are included/excluded? How much of the mean annual evaporation is taken into account when considering only the dry canopy conditions?

R: The main message of Fig. 4 is to show the model, with single parameter set, can reasonably well describe the mean and variability of daily evaporation across boreal sites in snow-free growing season, in absence of recent precipitation. As data from different sites does not represent same period (and length of dataset is different), adding diagnostics of dry vs. wet day distribution will add little to the present study.

The partitioning of ET into transpiration, evaporation from the ground and intercepted rainfall / sublimation of show on the canopy depends strongly on LAI, weather conditions (potential evaporation, distribution and intensity of rainfall) and time period considered. To elaborate this point in more detail, the ET data in Fig. 9 is in plotted against annual maximum LAI. By definition, focusing to dry-canopy conditions only would eliminate evaporation E and consequently 10 – 40% of annual evapotranspiration.  The contribution is likely smaller during growing season but strong dependence on precipitation frequency and intensity is expected.

[Figure]

Fig: Partitioning of long-term (2006-2016) evapotranspiration at C3 into components: Tr – transpiration; Ef – evaporation from the ground; E – evaporation / sublimation from wet canopy.

C: P34 Fig 7: Please spell out IQR. Should the right axis of subplot b be blue? Please add units.

R: Changed as suggested

C: P33Fig 6: Please consider including a measure of the runoff performance? And present the measure for all catchments?

R: The value of Willmott's index of agreement (eq. 22) was added to caption. Please note that same measure is given for all catchments in Supplementary Table 2.

C: Fig 35: Please state the catchment name considered.

R: Changed as suggested

C: P38: Please provide justification, explanation or reference for of all values selected.

R: Changed as suggested

**Response to Reviewer 2 comments**

We thank the Reviewer 2 for thorough comments and naturally share his/her opinion that the study provides important contribution and was worth doing. Relatively minor changes to manuscript structure are in our opinion sufficient to significantly improve clarity and address the important points raised by the reviewer. The main criticism was directed to some specific sections in the manuscript, where the structure and clarity could be significantly improved. In this respect we have received two somewhat conflicting reviews and have asked Editor Opinion on further changes.

**Specific comments (C) and our responses (R)**

C: It seems the authors have a chaotic design for the m/s, for example when Fig 3 is talked about in page 12 the authors suddenly refers to Fig 10, and then jump back to Fig 4

R: Referring Fig10 here was a mistake; should have been lowest panel of Fig3.

C: Page 4 Line 25 and 28: in the two equations, two symbols for the same leaf area index are used.

R: The leaf-area index (a stand-level property) is throughout the manuscript referred as LAI (m2m-2); the L (m2m-2) in the inline equation at P4L25 describes cumulative leaf-area index starting from the canopy top as clearly described in the manuscript. The relation of these variables is thus $LAI = \int_{hc}^{0} L(z)dz$ , where integration with depth z is from canopy top (z=hc) to ground (z=0).

C: Page 5 in the eq.6, describe how the threshold parameter is determined, based on measurements or calibrated?

R: As stated at P5L13-14 the soil moisture response and threshold parameter value was adopted from Lagergren and Lindroth (2001) sap-flow study. This part will be clarified in revised version.

C: Line 20: since fs is an important stress function in eq.4, it would be better to explicitly write it out, otherwise the readers have to find it by themselves in the literature cited. Note also the math express: at some places you used the form of y= exp(x), while at others you used y=ex. Be consistent.

R: Same addition was asked by Reviewer 1 and now added in the current manuscript.

C: Page 7: symbol beta is used in both eq.16 and eq.19, but with different meanings.

R: Thanks for pointing this out; term tan β in eq. 19 is now replaced with tan α.

C: Page 9 Line 8-10: delete – it is a repetition of page 8 Line 18-20. Also write in the 2.6.2 section clearly what a larger mu and sigma infer, otherwise when it comes to Results 3.1 and the relevant table, it is hard to comprehend.

R: Explanations what large mu and sigma intuitively mean are added and improve the clarity. We do not see overlap / repetition between P9L8-9 and P8L18-20

C: Page 11: The section 2.6 is 'Model validation at stand and catchment scales', but 2.6.3 is GIS preprocessing, and 2.6.4 is calibrating Topmodel. I don't think they are appropriately positioned. 2.6.3 is better fitted in Model input section, and 2.6.4 merged with parameter sensitivity section.

R: The model is tested/validated both at stand and catchment scales, and further sensitivity analysis is done at stand-scale (section 2.5.). To further improve the logic, we change the structure to be:

2.6 Model validation at stand scale; this includes the former section 2.6.1

2.7 Model validation at catchment scale; this includes the former section 2.6.2 and following sub-sections:

2.7.1 Processing of GIS-data

2.7.2 Calibration of Topmodel against specific discharge

We see this structure most logical for the manuscript content. The Model inputs –section (2.4) describe model inputs in general, and required for both spatial scales; Parameterization and sensitivity analysis (done at stand scale) is described in detail in 2.5, while the two validation scales are now in separate sections. We don't include Calibration of Topmodel into 2.5 as it is not part of sensitivity analysis and applies only to catchment scale simulations.

C: Page 14: paragraph starting from Line 4 can be written better by reorganizing a few sentences. The last sentence is important but the plot is not shown. Better show at least an example in one sub-slot either within the plot or next to the plot in Fig 6. Since this will be an important support for your argument.

R: We add a figure showing daily specific discharge at several catchments into Supplementary material (Fig. S2); these correspond to different 'goodness of fit' values based on the objective function value for discharge. Note also that objective function values are already given in Supplementary Table 1 for all catchments.

C: Page 14: 3.4.1- This part does not read well. I see you want to explain the temporal and spatial variability of soil moisture, and relate the variability with drainage and/or ET and its components. However, the current writing mixes them badly. Suggestion: describe Fig 7 well first for temporal variability, and Fig 8 second for spatial variability. For Fig 9 because you talk about it in the next section, so I suggest not to mention it here. Just describe the plots. All explanations can be moved to Discussion.

R: Thanks for good suggestions to improve the flow of the section. Referring to several figures is necessary to refer to specific processes and drivers of soil moisture variability. To improve the clarity and readability according to reviewer comments we make following changes:

1) New Fig 6: We merge Fig 6 and upper panels of Fig. 7; specific discharge, precipitation and snow water equivalent timeseries are now shown at top and temporal variability of mean soil moisture and its spatial variance at bottom. At both planes, x-axis is time.
2) New Fig 7: The lower panels of Fig 7 (mean soil moisture – spatial variance –planes) now form a separate figure.
3) We eliminate referring to multiple figures whenever possible and modify the text for clarity.

C: Line 27-28: Fig 9 is the long-term averaged values, cannot tell anything in between these years, like doy 180.

R: agree, reference to Fig. 9 removed

C: Line 30 – to support the rainfall effect, you must plot the rainfall bars in Fig 7.

R: see response above; Figs 6 & 7 were modified

C: Page 15: paragraph about SWE - move SWE to the last paragraph, i.e. describe fig 9 before fig 10, avoid mix them for mind-jumping

R: Good point, changed accordingly.

C: Page 16: Discussion – it is difficult to follow and digest, simply because it was badly organized. Currently it gives the impression of no logic of flow. Confusion is caused about what the main points are under discussion. Apparently, the proposed model has the capability to simulate hydrological processes across stand and catchment scales, and sensitivity analysis shows it only has a limited number of parameters significantly influencing the modeling results; Open GIS database application in hydrological model is also mentioned and discussed. Some features that the model has not developed are mentioned in Discussion which I think is unnecessary; and the reason why spatial validation of moisture, ET or SWE should be mentioned in Method validation section, not here. In one word, I would like the authors to think carefully about what they want to discuss or what are the main take-home messages they want to readers to get?

R: The reviewer has correctly found the main points of the manuscript; we tried to improve the revised version so that take-home message is more obvious. We shortened discussion on potential future developments and, and divide this section it into three subsections. 'Modeling ET at stand and catchment scales' addresses the first aim of the study to propose a new ET model for boreal landscape. The latter subsections 'Capabilities and limitations of the model framework' and 'Potential applications' discuss the proposed semi-distributed modular framework and its potential uses.

C: Page 18: first paragraph in 4.1.1 can be deleted. Part of the potential applicability of the model has already been mentioned in the previous paragraphs.

R: No actions taken; the Reviewer 1 suggested extending this section.

C: I hope the authors notice that there are many NOT SHOWN in the m/s, which makes the m/s sound incomplete.

R: thanks for noting; the revisions now eliminate this.

[revised manuscript text omitted]
 | 1.8 (0.4) | 1.7 | 12 (1) | 709 | 2.9 | 2002 | medium texture | Scots pine mineral soil | Kolari et al. (2004) |
| FISod | Sodankylä | 67.36N, 26.64E | 2.2 (0.2) | 13 (1) | 150 (50) | 527 | -0.4 | 2001-2009 | coarse texture | Scots pine mineral soil | Aurela (2005), Thum et al. (2007) |
| FIKal | Kalevansuo | 60.65N, 23.96E | 2.5 (0.5) | 15 (1) | 40 (3) | 606 | 4.6 | 2005-2008 | peat | Drained peatland | Lohila et al. (2011) |
| FILet | Lettosuo | 60.64N, 23.96E | 6.6 (2.3) | 20 (2) | 40 (3) | 627 | 4.6 | 2010-2012 | peat | Drained peatland | Koskinen et al. (2014) |
| SENor | Norunda | 60.09N, 17.48E | 6.5 (1.3) | 27 (3) | 100 (20) | 527 | 5.5 | 1996, 1997, 2003 | medium texture | Mixed coniferous, mineal soil | Lundin et al. (1999) Lindroth et al. (2008) |
| SEKno | Knottåsen | 61.0N, 16.22E | 3.6 (0.2) | 16.5 (1) | 40 (3) | 613 | 3.4 | 2006, 2009 | medium texture | Norway spruce mineral soil | Berggren et al. (2004), Lindroth et al. (2008) |
| SESky2 | Skyttorp 2 | 60.13N, 17.84E | 5.3 (0.5) | 15.8 (1) | 39 (2) | 527 | 5.5 | 2005 | medium texture | Scots pine mineral soil | Gioli et al. (2004) |
| FISii | Siikaneva | 61.85N, 24.30E | 0.3 (0.3) | 0.3 | | 709 | 2.9 | 2011, 2013, 2015 | peat | Boreal fen | Alekseychik et al. (2017) |

$LAI$ is ecosystem 1-sided leaf-area index (deciduous $LAI$ in parenthesis); $P$ is mean annual precipitation; $T_a$ mean annual air temperature and soil type refers to Table 2 in the main document.

* for runs at FIHy, site-specific field capacity $\theta_{fc} = 0.3$ and wilting point $\theta_{wp} = 0.1$ corresponding to main root zone (Launiainen et al., 2015) were used.

**Table 3.** Parameters and their ranges used in the global sensitivity analysis (Morris method) at stand scale.

| canopy parameters | range | unit | explanation |
|---|---|---|---|
| $LAI$ | 0.1 - 8.0 | $m^3\ m^{-3}$ | total leaf area index |
| $f_d$ | 0.0 - 1.0 | - | deciduous fraction |
| $g_1$ | 1.0 - 7.0 | - | stomatal parameter |
| $A_{max}$ | 6.0 - 14.0 | $\mu mol\ m^{-2}$ (leaf) $s^{-1}$ | maximum leaf net assimilation rate |
| $b$ | 20.0 - 60.0 | $W\ m^{-2}$ | half-saturation PAR of light response |
| $k_p$ | 0.4 - 0.6 | - | radiation attenuation coefficient |
| $r_w$ | 0.05 - 0.50 | - | critical relative extractable water |
| $G_f$ | $1.0\times10^{-3}$ - $1.0\times10^{-1}$ | $m\ s^{-1}$ | surface conductance for evaporation from wet forest floor |
| $h_c$ | 1.0 - 30.0 | $m$ | canopy height |
| $w_{max}$ | 0.5 - 3.0 | $mm\ LAI^{-1}$ | canopy storage capacity for rain |
| $w_{max,snow}$ | 1.0 - 10.0 | $mm\ LAI^{-1}$ | canopy storage capacity for snow |
| bucket parameters | | | |
| $z_s$ | 0.2 - 0.7 | $m$ | root zone depth |
| $z_{s,org}$ | 0.02 - 0.1 | $m$ | organic layer depth |
| $\theta_{fc,org}$ | 0.2 - 0.4 | $m^3\ m^{-3}$ | field capacity of org. layer |
| $\theta_{crit,org}$ | 0.1 - 0.4 | $m^3\ m^{-3}$ | critical vol. water content of org. layer |

**Table 4.** Sensitivity of *Canopy* sub-model predictions to parameter variability. Mean ($\mu$) and standard deviation ($\sigma$) of the distribution of elementary effects for evapotranspiration ($ET$), transpiration ($T_r$), evaporation from canopy interception ($E$), ground evaporation ($E_f$), and annual maximum snow water equivalent ($SWE$). Negative sign of $\mu$ indicate output variable decreases when parameter value increases. Units are in mm a$^{-1}$ except for $SWE$ (mm).

| Parameters | $ET$ $\mu$ | $ET$ $\sigma$ | $T_r$ $\mu$ | $T_r$ $\sigma$ | $E$ $\mu$ | $E$ $\sigma$ | $E_f$ $\mu$ | $E_f$ $\sigma$ | $SWE$ $\mu$ | $SWE$ $\sigma$ |
|---|---|---|---|---|---|---|---|---|---|---|
| $LAI$ | 100 | 230 | 240 | 280 | 140 | 94 | -130 | 77 | -36 | 33 |
| $f_d$ | -9.5 | 13 | -20 | 16 | -32 | 14 | 11 | 8.3 | 23 | 21 |
| $g_1$ | 97 | 82 | 97 | 82 | 0.0 | 0.0 | -0.0 | +0.0 | 0.0 | 0.0 |
| $A_{max}$ | 56 | 38 | 56 | 38 | 0.0 | 0.0 | -0.0 | 0.1 | 0.0 | 0.0 |
| $b$ | -42 | 23 | -42 | 23 | 0.0 | 0.0 | +0.0 | +0.0 | 0.0 | 0.0 |
| $k_p$ | -3.3 | 13 | 20 | 19 | 0.2 | 0.1 | -23 | 15 | -0.2 | 0.1 |
| $r_w$ | -7.3 | 6.6 | -7 | 6.6 | 0.0 | 0.0 | 0.0 | 0.0 | 0.0 | 0.0 |
| $G_f$ | 17 | 52 | -2.2 | 5.9 | 0.0 | 0.0 | 19 | 53 | 0.0 | 0.0 |
| $h_c$ | -8.2 | 29 | -9.0 | 2.0 | 20 | 24 | 0.7 | 13 | -4.4 | 4.9 |
| $w_{max}$ | -22 | 22 | -19 | 23 | 69 | 43 | -2.7 | 2.3 | 5.7 | 4.4 |
| $w_{max,snow}$ | -0.9 | 2.0 | -1.9 | 2.3 | 11 | 14 | 1.0 | 0.8 | -28 | 21 |
| $z_s$ | 58 | 44 | 58 | 44 | 0.0 | 0.0 | -0.1 | 0.3 | 0.0 | 0.0 |
| $z_{s,org}$ | 19 | 24 | -3.2 | 5.6 | 0.0 | 0.0 | 22 | 24 | 0.0 | 0.0 |
| $\theta_{fc,org}$ | 7.8 | 8.5 | -1.1 | 2.0 | 0.0 | 0.0 | 8.9 | 8.5 | 0.0 | 0.0 |

---

## Author Response (AR2)

hess-2019-45 / response to Editor comments

The manuscript has been significantly improved. Most points are addressed well, nonetheless some improvements still should be made:

* Title: I think i would be good to add 'boreal and peatland ecosystem to your title. Although you claim the model is also suitable for other ecosystems, you don't have any proof on this. Currently, the model is only tested for a boreal and peat land ecosystem.

R: changed accordingly

* I think the manuscript would benefit from an English language check. Especially focus on articles.

R: the manuscript was checked and numerous mistakes in articles corrected.

* P4 L20: the units of the model are mm/day! Not mm

R: did not find this mistake in the manuscript file; will be corrected in proof-reading phase if necessary.

* Eq 1: Why do the authors suddenly use E for evaporation? Before they used ET. Furthermore, I recommend removing the dt, since you define ET in mm/day. And rho_w should be in the denominator.

R: We have defined ET as evapotranspiration while in eq. 1 we want to emphasize that we talk about different ET components (E = evaporation from wet canopy, Tr = transpiration, Ef = evaporation from ground).

The typo in eq. 1 was corrected and rho_w put into denominator. Having dt in eq.1 is necessary to convert from mm/s to mm/d; this was added after previous comment.

* P5L1: the unit of L is J/kg

R: thanks for pointing out the typo; corrected

* P5L20: exp should not be in italic. This should be changed in throughout the manuscript. This also holds for ln, min, and max in equation form.

R: changed accordingly throughout the manuscript